# MermaidFlow: Redefining Agentic Workflow Generation via Safety-Constrained Evolutionary Programming

## Abstract

Despite the promise of autonomous agentic reasoning, existing workflow generation methods frequently produce fragile, unexecutable plans due to unconstrained LLM-driven construction. We propose **MermaidFlow**, a framework that redefines the agentic search space through safety-constrained graph evolution. At its core, MermaidFlow represent workflows as a verifiable intermediate representation using Mermaid, a structured and human-interpretable graph language. We formulate domain-aware evolutionary operators, i.e., *crossover*, *mutation*, *insertion*, and *deletion*, to preserve semantic correctness, enabling efficient exploration of a high-quality, statically verifiable workflow space. Without modifying task settings or evaluation protocols, MermaidFlow achieves consistent improvements in success rates and faster convergence to executable plans on the agent reasoning benchmark. The experimental results demonstrate that safety-constrained graph evolution offers a scalable, modular foundation for robust and interpretable agentic reasoning systems.

## 1 Introduction

Large language models (LLMs) are increasingly instantiated as modular agents that collaborate to solve complex tasks through structured workflows (Guo et al., 2024; Li, 2025a;b). These agentic workflows decompose problems into subtasks, assign them to specialized agents, and integrate intermediate outputs toward a shared goal. Moving beyond single-agent prompting, this multi-agent setting requires coherent planning and execution across agents with distinct roles and responsibilities. Designing such workflows involves reasoning over compositional graph structures that represent inter-agent dependencies, data flow, and semantic constraints, forming the foundation for scalable and adaptive multi-agent systems (Zhou et al., 2025).

The lifecycle of agentic workflow is naturally structured into three layers: (1) *workflow planning*, which defines the structure of subtasks, agent roles, and information flow; (2) *code realization*, where the plan is translated into executable programs; and (3) *runtime execution*, where agents are instantiated and carry out their assigned behaviors. In many systems, these layers are collapsed (e.g., Hu et al. (2024); Zhang et al. (2024c)): workflows are directly generated as *Python code* or serialized *JSON trees*, where planning decisions are entangled with implementation (i.e., through *prompting*-based generation of code or execution traces). As a result, workflows are often encoded in *low-level* formats where **structure** is implicit, **semantics** are entangled with imperative logic, and **validity** can only be assessed at runtime. This implicit representation hinders verifiability, reuse, and search, limiting the robustness and scalability of multi-agent systems.

Indeed, recent studies reveal that multi-agent LLM systems frequently fail due to brittle workflow logic and coordination breakdowns (Cemri et al., 2025; Zhang et al., 2024a; 2025c). These failures typically arise not from deficiencies in language models themselves but emerge from workflows that cannot be reasoned about, verified, or adapted. Without a structured representation of agent roles, task flow, and dependencies, systems struggle to detect errors before execution or to generalize behaviors across tasks. This points to a core limitation: **existing workflows lack the abstraction needed for reliable planning.**

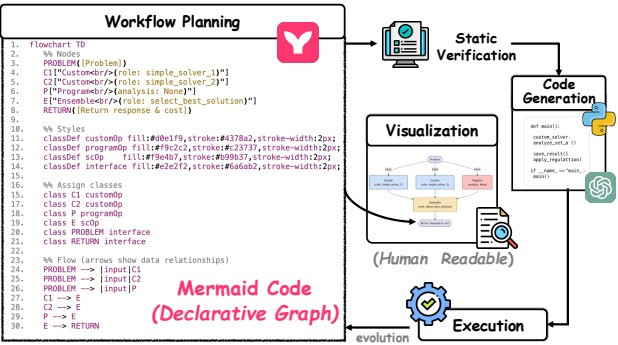

Figure 1: An illustration of the **workflow lifecycle in MermaidFlow**. The workflow is modeled as a declarative graph using Mermaid code, where nodes $\mathcal{V}_{[\tau,\alpha]}$ and edges $\mathcal{E}_{[\rho]}$ are explicitly defined with annotated prompts and roles (lines 3-8), styled and typed (lines 11–21), and connected via directed edges (lines 24–30). This results in a statically verifiable, semantically typed, and structurally interpretable representation that serves as a unified interface for visualization, validation, and code generation.

To address the limitations of implicit, code-bound workflows, we introduce **MermaidFlow**, a declarative representation for agentic planning inspired by the **Mermaid graph markup language**[1]. MermaidFlow defines workflows as **declarative graphs**, where nodes represent prompting agents and edges specify information flow (see Figure 1 for an illustrative example of the declarative graph encoded in Mermaid language, which is declarative, structurally explicit, and highly human-interpretable). This high-level representation enables structural and semantics properties with static verification, e.g., *structure feasibility*, and *type-safe connections*, can be enforced at the graph level, offering a clear plan that is both **human-readable** and **programmatically analyzable**. By exposing explicit semantics and structure, MermaidFlow yields substantial downstream benefits in both workflow generation and evaluation, when LLMs are extensively employed to discover and evaluate workflows. These properties ultimately yield a more robust, and verifiable space for agentic workflow planning. MermaidFlow enforces a clear separation between symbolic planning and executable code, ensuring that workflow structures remain statically verifiable by design.

Building on this foundation, we further propose a novel **evolutionary programming** (EP) framework tailored specifically to explore MermaidFlow's structured graph space. Our EP approach employs safety-constrained operations, including node replacement, subgraph rewiring, and role-consistent insertions, to maintain workflow correctness throughout the search process. Furthermore, historical workflows generated during search accumulate as structured experience, enabling efficient reuse and adaptation across tasks. Together, MermaidFlow's declarative representation and EP search framework constitute a programmable and task-agnostic programming layer for agentic workflow generation, enabling efficient search with improved correctness, generalization, and adaptability. To our knowledge, this is the first agentic workflow framework to **guarantee static graph-level correctness across the entire generation process.**

In summary, our contributions are threefold: (1) We introduce MermaidFlow, a declarative, verifiable graph representation for agentic workflow planning that cleanly separates planning from execution; (2) We develop a novel EP-based search framework leveraging structured mutation operators and workflow experience accumulation; and (3) We empirically demonstrate that MermaidFlow significantly outperforms existing code-based methods on standard agentic reasoning benchmarks, improving success rates, search efficiency, and interpretability.

## 2 RELATED WORKS

**Agentic Workflows with LLMs** Recent advances in multi-agent LLM systems have enabled structured collaboration among specialized agents to tackle complex, multi-step tasks. AFLOW (Zhang et al., 2024c), MaAS (Zhang et al., 2025b), and MASS (Zhou et al., 2025) formalize agent workflows using execution graphs and message-passing protocols to model multi-step reasoning. MetaGPT (Hong et al., 2024) and MAS-GPT (Ye et al., 2025) implement role-based orchestration by assigning domain-specific functions (e.g., product manager, engineer) and encoding Standard Operating Procedures (SOPs) to reduce cascading errors. Debate-based frameworks such as Multi-Agent Debate (Liang et al., 2024; Du et al., 2024) and DebFlow (Su et al., 2025) introduce structured critique among agents to promote output reliability. While these systems improve modularity and

---
[1]https://mermaid.js.org

scalability, their workflows are typically encoded in *imperative code* or *loosely structured prompts*, i.e., formats that lack semantic abstraction and resist verification. Recent studies (Cemri et al., 2025; Zhang et al., 2024a; 2025c) identify fragile workflows, rather than model errors, as the primary source of failure in multi-agent systems. Our proposed MermaidFlow addresses this bottleneck by introducing a typed, declarative workflow space that supports safe construction, static validation, and structured exploration, advancing agentic reasoning beyond brittle prompt chaining.

**Workflow Representation** The representation of agentic workflows governs not only how agents are composed, but also whether they can be verified, reused, or optimized. Natural language-based prompting methods, such as Chain-of-Thought (Kojima et al., 2022), ReAct (Yao et al., 2023), and Self-Refine (Madaan et al., 2023), are expressive but underspecified, lacking formal structure for validation. In contrast, code-centric approaches like AFLOW (Zhang et al., 2024c), ADAS (Hu et al., 2024), and ScoreFlow (Wang et al., 2025) generate executable Python or JSON trees directly, offering precision at the cost of brittleness and poor editability due to tightly entangled logic and implementation. Recent efforts explore more structured workflow abstractions. GPTSwarm (Zhuge et al., 2024) and FlowReasoner (Gao et al., 2025) organize workflows as agent interaction graphs, but lack formal semantics, e.g., no type enforcement, role validation, or support for systematic search. MetaGPT (Hong et al., 2024) and MAS-GPT (Ye et al., 2025) encode workflows through SOP-style templates and DSLs, but rely on rigid decomposition patterns that restrict flexibility. MermaidFlow departs from these by introducing a typed, declarative graph representation grounded in the Mermaid markup language. It makes role semantics and data flow explicit, allowing crucial graph-level structural constraints, such as role consistency, and type safety, to be enforced pre-execution, enabling safe reuse, adaptation, and search.

**Workflow Search and Optimization** The structure of the search space fundamentally shapes how workflows are generated and optimized. AFlow (Zhang et al., 2024c) applies Monte Carlo Tree Search over executable graphs, while ADAS (Hu et al., 2024) explores code-level candidates via heuristics-guided expansion. Though systematic, both approaches operate over brittle code-centric representations, where small mutations often break correctness, necessitating expensive filtering. ScoreFlow (Wang et al., 2025) and G-Designer (Zhang et al., 2024b) adopt learned or continuous optimization strategies, adjusting prompt topologies or agent graphs via gradient-based tuning or neural controllers. However, these methods require differentiable feedback or training signals and offer limited support for enforcing structural validity. A complementary direction leverages evolutionary and population-based search. DebFlow (Su et al., 2025) refines workflows through iterative agent debates, while EvoFlow (Zhang et al., 2025a) evolves diverse workflows using task complexity-conditioned genetic search. Yet both approaches operate in loosely defined or weakly constrained spaces, where mutations often yield semantically invalid workflows. MermaidFlow closes this gap by introducing a structured, verifiable graph space equipped with safety-aware mutation operators. This design guarantees that every candidate is valid by construction, enabling scalable and principled workflow optimization.

# 3 A NOVEL DECLARATIVE GRAPH REPRESENTATION FOR AGENTIC WORKFLOWS

This section introduces a declarative graph representation for agentic workflows, built on **Mermaid** that is a structured, human-readable language with built-in static verifiability and graph render to help human directly observe the workflow. Departing from unstructured or token-level workflow representations, our workflow formalism leverages Mermaid's type-aware syntax to enable correctness by construction, symbolic manipulation, and modular workflow composition.

## 3.1 DECLARATIVE WORKFLOW GRAPHS WITH MERMAID

We model each agentic workflow as a declarative computation graph with explicit typing, annotations, and semantic structure. Formally, we define a workflow graph as:

$$G(\mathcal{V}_{[\tau,\alpha]}, \ \mathcal{E}_{[\rho]}), \tag{1}$$

where $\mathcal{V}_{[\tau,\alpha]}$ is a set of typed and annotated nodes, $\mathcal{E}_{[\rho]}$ is a set of directed, role-labeled edges.

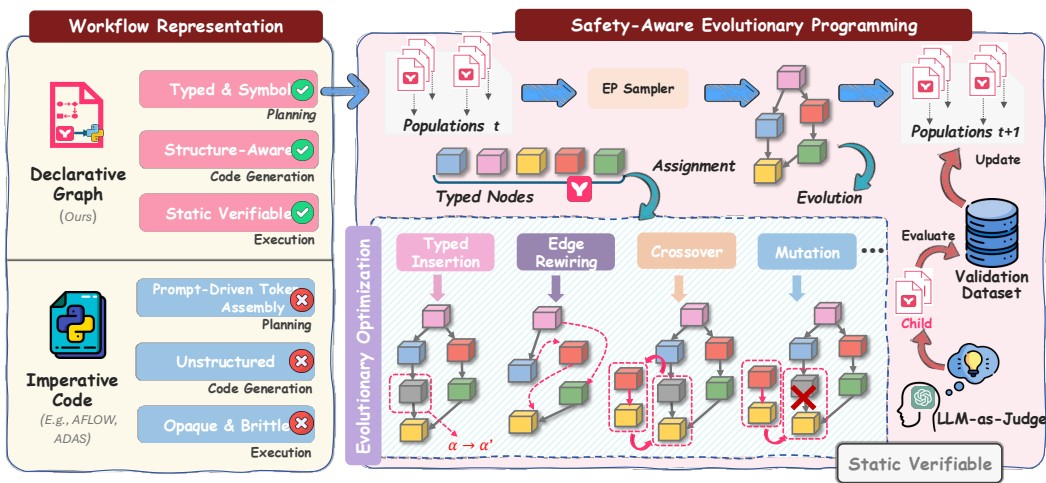

Figure 2: **Overview of the MermaidFlow framework**. **Left**: Comparison between *imperative* (Python-based) and *declarative* (Mermaid-based) workflow representations. MermaidFlow models workflows as statically typed, verifiable graphs, enabling interpretable planning and structure-aware code generation. **Right**: Illustration of the safety-aware evolutionary programming process. Given historical Mermaid workflows, the EP sampler selects parent candidates and applies EP operators to generate new workflow candidates. An LLM-as-Judge then selects the final workflow for evaluation, and the results are used to update the population.

This structure is instantiated using the **Mermaid** graph language, a lightweight, human-readable syntax for specifying typed graphs with semantically annotated components. Furthermore, Mermaid provides a declarative interface that supports symbolic manipulation and static validation. A real example is illustrated in Figure 1. Each node defines a symbolic identifier and type signature, while edges carry semantic labels (e.g., `inputs`) that describe data-flow.

Next, we define each component of the workflow in the Mermaid domain to illustrate how the workflow aligns well with its Mermaid representation.

**Nodes.** Each node $v \in \mathcal{V}_{[\tau, \alpha]}$ is a tuple (`id`, $\tau(v)$, $\alpha(v)$), where `id` is a symbolic identifier, $\tau(v) = \mathtt{T_{in}} \to \mathtt{T_{out}}$ denotes the type of that node, and attribute $\alpha(v)$ would provide some necessary information according to the type. As shown in lines 4–7 of fig. 1, each node element can be concisely defined in a single line of Mermaid script, for example: (id: C1, type $\tau$: CustomOp, attribute $\alpha$: role-simple_solver_1). Nodes represent typed declarative units that are interpretable and statically verifiable, and it can be easily understood by human.

**Edges.** Each edge $(u, v) \in \mathcal{E}_{[\rho]}$ denotes a dependency annotated with a semantic label $\rho(u, v)$, indicating how information or control flows from $u$ to $v$ e.g., "input", "problem". Mermaid syntax supports these semantics with labeled arrows (e.g., `A -> |input| B`).

**Types.** All graph nodes are explicitly typed and semantically annotated, with types governing interface compatibility and ensuring valid workflow construction. By defining these types up front, we guarantee a consistent translation from Mermaid diagrams to Python code. For each task domain, we introduce dedicated node types for the operators and tools that have proven most effective. During code generation, these attributes direct the translator to emit the correct Python calls, making sure that all tool arguments are clearly specified. We provide a detailed type description in Appendix A.1.

Each node explicitly defines its symbolic identity, type signature, and related attribution, while edges govern execution flow subject to type constraints. Unlike flat or post-validated node-link DAGs (e.g., JSON plans or token-generated programs), our declarative graph formalism introduces an abstraction layer that bridges symbolic reasoning and execution-level safety. This structure supports (static) correctness-preserving mutation and compositional reuse, which are the key properties we exploit in the constrained optimization process detailed next. To the best of our knowledge, this is the first agentic workflow representation that leverages a graph-oriented abstract coding language to enable more natural graph definition and manipulation. In the next section, we will formalize graph-

manipulation actions in Mermaid and present a workflow-optimization method, further illustrating the advantages of using Mermaid for workflow representation.

## 3.2 Agentic Workflow Search Space

The declarative graph formalism introduced above induces a constrained search space over **agentic workflows**. We define the workflow together with related LLM factors as follows:

$$\mathcal{S} = \left\{ G(\mathcal{V}_{[\tau,\alpha]}, \ \mathcal{E}_{[\rho]}, \ \mathcal{C}) \in \mathcal{G}_{\text{Mermaid}} \ \big| \ G \models \mathcal{C}_{\text{static}} \right\}, \tag{2}$$

where $\mathcal{G}_{\text{Mermaid}}$ denotes the set of workflows expressible in the Mermaid graph language, and $\mathcal{C}_{\text{static}}$ captures structural constraints, such as type compatibility, role-consistent edges, and connectivity, automatically enforced by Mermaid's parser and extended structural schema. This built-in verifiability arises from Mermaid's declarative syntax and ensures that all elements of $\mathcal{S}$ are valid and executable by construction.

To enable optimization, we parameterize each node $v \in \mathcal{V}_{[\tau,\alpha]}$ as a tuple $v = \big( m, \ p(\tau,\alpha), \ f(\tau) \big)$, following conventions from multi-agent systems (MAS) definition, where $m \in M$ specifies the LLM configuration (e.g., `model_name`, `temperature`), $p(\tau,\alpha) \in P$ is the prompt template determined by the node type $\tau$ and its argument $\alpha$, and $f(\tau) \in F$ denotes the input/output format associated with type $\tau$.

$$\mathcal{V}_{[\tau,\alpha]} = \{(m, p(\tau,\alpha), f(\tau) \mid m \in M, \ p \in P, \ f \in F\}. \tag{3}$$

The formula above demonstrates that, when interpreting a Mermaid workflow, each node can be directly mapped to a standard LLM agent instance. By assigning a type $\tau$ and parameter/attribute $\alpha$, one can associate the node with a specific prompt $p(\tau,\alpha)$ and input/output format $f(\tau)$. **This formulation emphasizes that every LLM agent can be consistently defined both within the Mermaid representation and in the general context of LLM agent configuration.**

The space is also **inductively closed**: type-compatible subgraphs can be composed without revalidation, while disconnected or cyclic fragments are excluded by construction. This closure property is non-trivial: prior workflow representations, especially those based on imperative or token-level programs, **lack structural guarantees**, and mutations frequently yield invalid or unrecoverable states. In contrast, our declarative design ensures that local edits remain within the valid region of $\mathcal{S}$, enabling reliable and efficient search. Though intentionally bounded, $\mathcal{S}$ remains expressive enough to capture planning motifs such as hierarchical refinement and dataflow composition. By unifying generation, mutation, and verification within a single, compiler-verifiable substrate, it provides a semantically grounded foundation for structure-aware and safe workflow optimization.

## 4 Constraint-Aware Evolutionary Workflow Optimization

We introduce an evolutionary programming (EP) framework that operates directly over declarative Mermaid workflows. Leveraging Mermaid's typed and verifiable graph structure, we define correctness-preserving operators that enable safe, modular workflow evolution. Unlike prior approaches over unstructured or token-based spaces, all candidates in MermaidFlow are valid by construction, ensuring safe, compiler-checkable optimization throughout the search process.

### 4.1 Constraint-Preserving EP Operators for Declarative Workflow Graphs

We define a set of atomic graph-level operators that drive workflow evolution within MermaidFlow. Each operator acts over a candidate graph $G(\mathcal{V}, \mathcal{E}) \in \mathcal{G}_{\text{Mermaid}} \subseteq \mathcal{S}$, and is designed to be *locally scoped*, *type-consistent*, and *statically verifiable*, enabling every candidate to be validated by the Mermaid compiler at each step of the search. Below are the definitions of the operations, which will be used to verify the correctness of the newly generated workflow.

**Node Substitution.** Changing the attributes of a specific agent $v(\tau, \alpha) \in \mathcal{V}$ to $v(\tau, \alpha\prime)$. Like changing the corresponding role prompt or instruction.

**Node Addition.** Given an edge $(v_a, v_b) \in \mathcal{E}$, connecting from node $v_a$ to node $v_b$, insert a new node $v\prime$ to form $(v_a, v')$ and $(v', v_b)$ and disconnect $(v_a, v_b)$ if: $\texttt{T}_{\text{out}}(v_a) = \texttt{T}_{\text{in}}(v'), \quad \texttt{T}_{\text{out}}(v') = \texttt{T}_{\text{in}}(v_b)$ according to their node type $\tau$.

**Edge Rewiring.** Given nodes $\{v_a, v_b, v_c\} \subseteq \mathcal{V}$ and $(v_a, v_b) \in \mathcal{E}$ in the original graph $G$, rewire to $(v_a, v_c)$ or $(v_c, v_b)$ and disconnect $(v_a, v_b)$ if: $\mathtt{T_{out}}(v_a) = \mathtt{T_{in}}(v_c)$ or $\mathtt{T_{out}}(v_c) = \mathtt{T_{in}}(v_b)$.

**Node Deletion.** Given a linear path $v_a \rightarrow v_b \rightarrow v_c$, delete $v_b$ and insert an edge $(v_a, v_c)$ if $\mathtt{T_{out}}(v_a) = \mathtt{T_{in}}(v_c)$.

**Subgraph Mutation.** Let $G_1 \in \mathcal{G}_{\text{Mermaid}}$ be a subgraph of the graph $G \in \mathcal{S}$. Denote the input and output node set of $G_1$ as $I_1$ and $O_1$, respectively. Let $G_2 \in \mathcal{G}_{\text{Mermaid}}$ be a feasible graph with input and output node set $I_2$ and $O_2$. Replace $G_1$ in $G$ with $G_2$ such that $\mathtt{T_{in}}(I_1) = \mathtt{T_{in}}(I_2)$ and $\mathtt{T_{out}}(O_1) = \mathtt{T_{out}}(O_2)$.

**Crossover.** Given $\{G_1, G_2\} \subseteq \mathcal{S}$ share a common interface node $v$ (e.g., an ensemble node), swap subgraphs rooted at $v$ to yield $\{G_1', G_2'\}$ such that the type and interface constraints are preserved, i.e., $\{G_1', G_2'\} \subseteq \mathcal{S}$. Each operator is applied at the level of Mermaid syntax, enabling compiler-level validation of every candidate graph. By constraining transformations to preserve type and role integrity, MermaidFlow ensures that evolutionary search remains within the semantically valid subspace of workflows. In the case study fig. 4, there is a concrete example illustrating the crossover operator.

**Lemma 1** (MermaidFlow Transformation Invariance). *Let $\mathcal{S}$ denote the declarative workflow space defined in Section 3.2. For any workflow graph $G \in \mathcal{S}$ and any atomic transformation operator $\mathcal{O}$ defined above, the resulting graph $G' = \mathcal{O}(G)$ also belongs to $\mathcal{S}$:*

$$\forall G \in \mathcal{S}, \ \forall \mathcal{O} \in \mathbb{O}, \quad \mathcal{O}(G) \in \mathcal{S} \tag{4}$$

*where $\mathbb{O}$ is the set of constraint-preserving operators over MermaidFlow graphs. That is, $\mathcal{S}$ is closed under all valid EP operations.*

**Definition 1.** *Let $\mathcal{G}$ denote the space of all candidate workflows, each $G \in \mathcal{G}$ represented as a directed graph $(\mathcal{V}, \mathcal{E})$. We define a static validator function $Q : \mathcal{G} \rightarrow \{0, 1\}$, implemented by a Mermaid parser/compiler, such that:*

$$Q(G) = \begin{cases} 1 & \text{if } G \in \mathcal{S} \\ 0 & \text{otherwise} \end{cases} \tag{5}$$

*Here, $\mathcal{S} \subset \mathcal{G}$ is the subset of workflows satisfying verifiability constraints such as workflow structure, well-typed I/O, role validity, and full connectivity.*

By using EP operators above, from Lemma 1, given a $G_t \in \mathcal{S}$, each change $\mathcal{O}(G_t)$ at step $t$ leads to a graph $G_{t+1} = \mathcal{O}(G_t) \in \mathcal{S}$. Given an initial graph $G_0 \in \mathcal{S}$, by induction, we know $\forall t \in \mathbb{N}_+, G_{t+1} = \mathcal{O}_t \circ \mathcal{O}_{t-1} \cdots \circ \mathcal{O}_0(G_0) \in \mathcal{S}$. Thus, the evolution in the static Mermaid graph space remains the safe subspace.

In MermaidFlow, when using an LLM to generate a new Mermaid graph, the resulting Mermaid code may sometimes violate predefined safety constraints. To address this, we implement a checker to verify whether the newly generated candidates conform to the defined workflow and operation rules. If any violations are detected, new workflows are regenerated. Thanks to Mermaid's simple and clear syntax, the code can be treated as structured text. This allows us to easily build a text-based analysis tool and incorporate custom rules into the checker. More implementation details can be found in Appendix A.2.

## 4.2 Evaluation and Selection in Workflow Populations

We frame each declarative workflow graph as an *experience* and maintain a population of scored experiences over time. At each optimization step $t$, the system tracks a history buffer: $W_{\text{history},t} = \{(s_i, \text{score}_i)\}_{i=1}^{t}$, where $s_i \in \mathcal{G}_{\text{Mermaid}}$ denotes a structurally valid workflow, and score$_i$ reflects its estimated performance.

At each optimization cycle, two parent workflows $s_a, s_b$ are sampled from $W_{\text{history},t}$, typically via temperature-scaled softmax sampling according to following distribution: $P_{\text{mixed}}(i) = \lambda \cdot \frac{1}{t} + (1 - \lambda) \cdot \frac{\exp(\alpha \cdot score_i)}{\sum_{j=1}^{n} \exp(\alpha \cdot score_j)}$, where $t$ is the number of workflows in the history buffer, $score_i$ is the validation score of the $i$-th workflow, and the parameters $\alpha$ and $\lambda$ control the influence of the scores, and

balances exploration-exploitation, respectively. After sample two different parent workflows $s_a, s_b$ where $s_a \neq s_b$. These are used to generate a candidate set through the evolutionary process:

$$S_{\text{candidates}} = \{s_i \mid s_i = \mathcal{O}(s_a, s_b), \ \mathcal{O} \in \mathbb{O}\}_{i=1}^N , \tag{6}$$

where $\mathbb{O}$ denotes the set of correctness-preserving operators (Section 4.1), for some operator only $s_a$ involved and $N$ is the candidate pool size.

To avoid expensive rollout-based evaluation over the full population, we adopt an *LLM-as-judge* model that scores each candidate $s \in S_{\text{candidates}}$ based on semantic fit, structure, and task relevance. Since all candidates in $S_{\text{candidates}}$ are statically verified by the Mermaid compiler, they are guaranteed to be syntactically valid, type-safe, and structurally executable, dramatically reducing failure cases and increasing effective sample quality.

We then select the highest-scoring candidate and update the history buffer:

$$W_{\text{history},t+1} \leftarrow W_{\text{history},t} \cup \left\{(s^*_{\text{child}}, \ \text{Validate}(s^*_{\text{child}}))\right\}, \ \text{where } s^*_{\text{child}} = \arg\max_{s \in S_{\text{candidates}}} \text{LLM\_as\_Judge}(s).$$

This experience-centric design, enabled by the declarative and verifiable structure of MermaidFlow, supports efficient, low-cost population evolution without compromising safety, correctness, or search quality. See Appendix A.3 for algorithmic details.

## 5 EXPERIMENTS

### 5.1 EXPERIMENT SETUP

**Baseline.** We choose threefold of agentic baselines: (1) **Non-agentic reasoning methods**, including CoT (Kojima et al., 2022), ComplexCoT (Fu et al., 2023), and Self-Consistency (Wang et al., 2023). (2) **Hand-crafted multi-agent systems**, such as LLM-Debate (Du et al., 2024), LLM-Blender (Jiang et al., 2023), DyLAN (Liu et al., 2024), and MAcNet (Qian et al., 2024). (3) **Autonomous multi-agent systems**, including GPTSwarm (Zhuge et al., 2024), MaAS (Zhang et al., 2025b), AutoAgents (Chen et al., 2024), ADAS (Hu et al., 2024), and AFlow (Zhang et al., 2024c). Among them, GPTSwarm and MaAS incorporate trainable modules for assigning workflow structures, while AutoAgents, ADAS, and AFlow rely on an LLM to design the structure, consistent with our setting. More details on baseline setups are provided in Appendix A.4.

**Task and Benchmarks.** We evaluate MermaidFlow on four public benchmarks covering two domains: **(1) math reasoning**, GSM8K (Cobbe et al., 2021), MATH (Hendrycks et al., 2021); **(2) code generation**, HumanEval (Chen et al., 2021), and MBPP (Austin et al., 2021). For MATH benchmark, we follow AFlow (Zhang et al., 2024c) and MaAS (Zhang et al., 2025b) in using the same selected problems from four typical problem types in level 5. The dataset statistics are provided in Appendix A.5.

**Implementation details.** We use a closed-source LLM (gpt-4o-mini-0718) as both the Optimization and Execution LLM, consistent with the setup in MaAS (Zhang et al., 2025b). The Optimization LLM is responsible for tasks such as generating promising workflows in Mermaid code, selecting from sampled workflows, evolving to new workflows, and translating Mermaid code into Python code. All models are accessed via API with the temperature set to 0. In each round, we generate four different $s_{\text{child}}$ candidates. To ensure experimental stability, complex operations such as crossover are applied with only a 10% probability. We set the number of iteration rounds to 20 for both Mermaid and AFlow, and to 30 for ADAS. The evaluation metrics are kept consistent with those used in AFlow and MaAS: for GSM8K and MATH, we report the Solve Rate (%) as the primary metric, while for HumanEval and MBPP, we report the pass@1 score.

### 5.2 EXPERIMENTAL RESULTS

We compare MermaidFlow against 13 baselines on the GSM8K, MATH, HumanEval, and MBPP benchmarks, as shown in Table 1. The results demonstrate that MermaidFlow consistently achieves the best performance across all tasks. Compared to methods that search for the next workflow in the Python field, such as ADAS and AFlow, our approach outperforms them by an average margin of 2.08% to 5.54%. On the MATH benchmark specifically, MermaidFlow exceeds the second-best method AFLOW by 2.61%. For certain benchmarks, performance is primarily limited by the

Table 1: Performance comparison among Non-agentic reasoning methods, hand-crafted multi-agent systems, and automated agentic workflows. All methods use `gpt-4o-mini` as the base LLM and are evaluated on the test split, with results averaged over three runs. **Bold** indicates the best result; underline denotes the runner-up. MermaidFlow shows consistent improvements across all datasets. *: Result reported in the MaAS paper, as the corresponding implementation for this dataset is not available in their code.

| Method | GSM8K | MATH | HumanEval | MBPP | Avg. |
|---|---|---|---|---|---|
| Vanilla | 87.57 | 46.29 | 87.49 | 70.29 | 72.91 |
| CoT (Kojima et al., 2022) | 87.45 | 46.40 | 88.13 | 71.83 | 73.45 |
| ComplexCoT (Fu et al., 2023) | 86.89 | 46.40 | 87.49 | 72.36 | 73.29 |
| SC (CoT×5) (Wang et al., 2023) | 87.57 | 47.91 | 88.60 | 73.60 | 74.42 |
| LLM-Debate (Du et al., 2024) | 89.47 | 48.63 | 88.80 | 70.29 | 74.30 |
| LLM-Blender (Jiang et al., 2023) | 88.35 | 46.92 | 88.68 | 77.05 | 75.25 |
| DyLAN (Liu et al., 2024) | 89.98 | 48.54 | 90.42 | 77.30 | 76.56 |
| MacNet (Qian et al., 2024) | 87.95 | 45.18 | 84.57 | 65.28 | 70.75 |
| GPTSwarm (Zhuge et al., 2024) | 89.14 | 47.88 | 89.32 | 77.43 | 75.94 |
| MaAS (Zhang et al., 2025b) | 91.47 | 52.19 | 91.57 | 82.17* | 79.35 |
| AutoAgents (Chen et al., 2024) | 87.69 | 45.32 | 87.64 | 71.95 | 73.15 |
| ADAS (Hu et al., 2024) | 88.35 | 43.18 | 84.19 | 77.05 | 73.69 |
| AFlow (Zhang et al., 2024c) | 90.11 | 52.81 | 90.08 | 81.67 | 78.67 |
| **MermaidFlow (Ours)** | **92.39** | **55.42** | **92.87** | **82.31** | **80.75** |

capabilities of the Execution LLM. For example, in HumanEval and GSM8K, the baseline (Vanilla) performance is already high, so improvements from architectural optimization are less significant. In contrast, for benchmarks where the baseline performance is relatively low, such as MATH and MBPP, our method demonstrates a more substantial impact. Overall, MermaidFlow's average score across these tasks is 80.75%, which is 1.40% higher than the highest average among all baselines (79.35% by MaAS), fully demonstrating the robustness and superiority of our approach across different problems.

## 5.3 ABLATION STUDY

**Evolution Efficiency** To evaluate the effectiveness of our approach, we compare the learning curves of MermaidFlow and AFlow on the MATH dataset, as shown in Figure 5.3. MermaidFlow demonstrates a more consistent improvement in workflow quality during training and better generalization to the test set.

The core difference between MermaidFlow and AFlow lies in the search space. AFlow operates directly on Python code, applying textual edits with prompts constraints. This approach often leads to invalid and non-functional programs, with only a 50% success rate in generating executable code. In contrast, MermaidFlow evolves workflows at the graph level using Mermaid, a domain-specific language that enables structured manipulation (e.g., adding, deleting, or mutating nodes). This

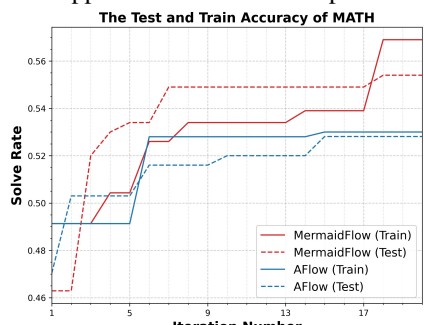

Figure 3: An illustrative figure comparing the highest solve rates on the MATH dataset between MermaidFlow and AFlow on the training set (119 problems) and test set (486 problems) across optimization iterations.

representation is better suited for LLM-based optimization (e.g., gpt-4o-mini), **consistently yielding >90% success rate in producing valid Python code**. This reliability enables more effective exploration and optimization of workflow space. Thanks to Mermaid's reliable generation rate and lightweight representation, it achieves better token efficiency. When AFlow and MermaidFlow both surpass 52% on the MATH dataset, they consume $6.9e4$ and $2.7e4$ tokens respectively, with MermaidFlow requiring only about half the cost of AFlow.

**Impact of Optimization LLM Scale** We investigate how the choice of Optimization LLM influences the quality of workflows in MermaidFlow by evaluating more capable models on the `HumanEval` and `GSM8K` benchmark. Specifically, we compare the effectiveness of larger models,

Table 2: Comparison of different optimization LLMs on `HumanEval` and `GSM8K` datasets.

| Dataset | Claude 3.5 | GPT-4o | GPT-4o-mini |
|---|---|---|---|
| HumanEval | 93.13 | 94.66 | 92.87 |
| GSM8K | 93.83 | 93.94 | 92.39 |

such as Claude 3.5 and GPT-4o, in generating new workflows, while keeping GPT-4o-mini fixed as the Execution LLM. Results are summarized in Table 2. As a result, higher-capacity optimization LLMs consistently yield better performance across both `HumanEval` and `GSM8K`. This consistent trend underscores a core strength of MermaidFlow: its well-structured, statically verifiable search space enables even modest improvements in optimization quality to translate directly into more functional, high-reward workflows.

**Optimal Stopping Point Analysis** We investigate the advantages of using Mermaid as the workflow representation in workflow update control. A stable and reliable search process requires controllable and well-defined update steps. With Mermaid, updates can be expressed through precise graph-based operations such as adding nodes, deleting nodes, or modifying edges. These structured operations help ensure that the newly generated workflow remains close to its parent workflow. In contrast, representing workflows directly in Python often restricts updates to vague instructions like "modify no more than five lines," which can lead to unreliable or semantically meaningless changes, causing the new workflow to deviate significantly from its parent.

We use the round index of optimal stopping points to demonstrate this.

Table 3: Final selected workflow indices for each benchmark.

| Method | GSM8K | MATH | HumanEval | MBPP |
|---|---|---|---|---|
| AFLOW | 8 | 15 | 5 | 8 |
| MermaidFlow | **16** | **18** | **7** | **10** |

These results show that MermaidFlow consistently discovers higher-quality workflows at later rounds, indicating a more stable and productive search trajectory compared to AFlow.

## 5.4 CASE STUDY: WORKFLOW EVOLUTION WITH MERMAIDFLOW

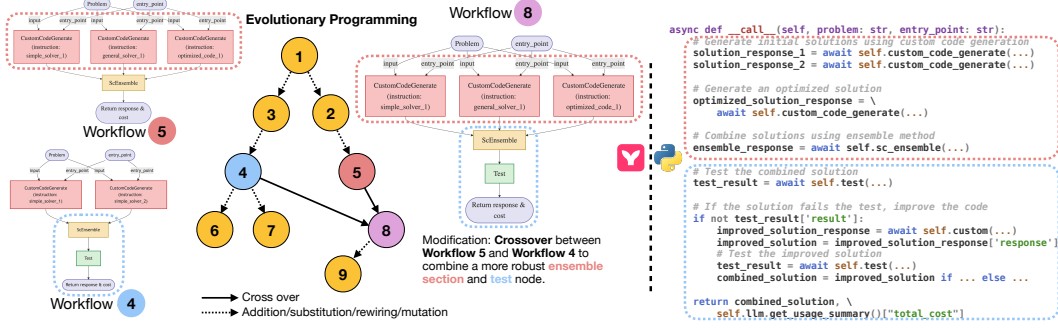

Figure 4: A case study on the `HumanEval` dataset showcasing how MermaidFlow evolves structured agentic workflows through evolutionary programming (with a detailed example of the *crossover* operator). The declarative graph representation also enables reliable translation of workflow graphs into *executable python code* (zoom-in view recommended).

In this case study, we present an example of how to generate a new workflow from given parent workflows using the Mermaid representation. During the evolutionary process, a new workflow can be derived from either a single parent workflow or two parent workflows, depending on the type of update operator applied. In the specific example shown in Figure 4, which is based on solving problems from the `HumanEval` benchmark, `Workflow_8` is generated based on `Workflow_4` and `Workflow_5`. Each parent contributes distinct advantages: `Workflow_4` is the first to

introduce a test node, while `Workflow_5` contains a more diverse ensemble section with agents covering different reasoning aspects. MermaidFlow combines these strengths to synthesize a new and improved `Workflow_8`.

This generation process occurs in the Mermaid Field, where all workflows are defined in a structured syntax that can be directly rendered as visual diagrams. Once a new Mermaid workflow is generated, we use `gpt-4o-mini` to translate the Mermaid code into executable Python code. Due to Mermaid's well-structured nature, this translation can be both straightforward and reliable. As demonstrated in Figure 4, the generated Python code perfectly resemble Mermaid `Workflow_8`, consisting of a diverse ensemble section and a test function.

This case study not only demonstrates the efficiency of searching for new high-quality workflow populations in the Mermaid field but also provides a detailed illustration of MermaidFlow's stable and composable workflow lifecycle.

## 6 CONCLUSION

We propose MermaidFlow, a framework that transforms agentic workflow generation by encoding workflows as statically typed, semantically annotated, and compiler-verifiable graphs using the Mermaid language. Our proposed workflow formulation defines a well-structured, declaratively defined search space that supports safety-constrained rewrites and modular composition. Building on this space, we develop a safety-constrained evolutionary programming framework that enables efficient, verifiable, and high-quality workflow synthesis. MermaidFlow offers a principled step toward structurally safer and more interpretable agentic systems, introducing the first workflow optimization framework built atop a statically verifiable workflow representation. While MermaidFlow is evaluated in controlled agentic reasoning settings, its integration with real-world multi-agent systems and user-in-the-loop workflows introduces nuances that merit further exploration.

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

## A  IMPLEMENTATION DETAILS

### A.1  DETAILS OF MERMAID

Mermaid is a text-based language for creating diagrams and flowcharts, which we use to represent workflows. In our system, we extend Mermaid with custom node types:

1. **CustomOp** - Used for specialized reasoning strategies, executing specific tasks through defined roles. Example: `K["Custom
(role:  validate_1)"]`, styled with blue fill.

2. **Interface** - Entry and exit nodes for workflows, including problem input points and return output points. Example: `PROBLEM([Problem])`, styled with light purple fill.

3. **ProgrammerOp** - Nodes that execute code generation and computation, particularly suitable for mathematical problems. Example: `P["Programmer
(analysis: 'Calculate step by step')"]`, styled with red fill. Only math problems use this node type.

4. **ScEnsembleOp** - Nodes that combine multiple solutions into one result, requiring multiple inputs to function properly. Example: `ENSEMBLE["ScEnsemble
"]`, styled with yellow fill.

5. **TestOp** - Nodes for validating solutions, used to verify whether code passes predefined test cases. Example: `T["Test
"]`, styled with green fill. Only code problems use this node type.

6. **CustomCodeGenerateOp** - Nodes specifically designed to generate code based on problem descriptions. Example: `CODE_GEN["CustomCodeGenerate
(instruction:  xxx)"]`, styled with red fill. Only code problems use this node type.

Each node type has specific connection rules and usage patterns to ensure data flows correctly through the workflow. Connections between nodes are typically represented by arrows, for example: `PROBLEM -> |problem| P`, indicating that the problem input is passed to the programmer node.

We provide detailed descriptions of each node type in our prompts to guide the LLM in generating structurally valid Mermaid workflows. Additionally, we implement a Mermaid checker to verify the correctness of the generated Mermaid code.

For different types of problems (math or code), we use different prompt templates to guide the LLM in generating appropriate Mermaid code, as each problem type involves different node types and workflow patterns.

For math problems, our templates emphasize clarity in computational steps and rigor in mathematical reasoning. For example, we encourage using Programmer nodes for precise calculations and `ScEnsemble` nodes to combine multiple solutions for improved accuracy. The prompt templates include guidelines for LaTeX mathematical expression formatting to ensure outputs conform to mathematical standards.

For code problems, our templates focus on code generation, testing, and validation. We particularly emphasize using Test nodes to validate generated code solutions and `CustomCodeGenerate` nodes to produce code based on problem descriptions. The prompt templates include guidelines for code formatting and testing requirements to ensure the generated code executes correctly and passes test cases.

We maintain a dedicated `prompt.py` module to store any node prompts that are too verbose for the Mermaid diagram, keeping the workflow clean and readable.

### A.2  MERMAID CHECKER IMPLEMENTATION

We implemented a comprehensive Mermaid validation system that performs both soft and hard checks on workflow diagrams. The soft check analyzes the structure using regex pattern matching to extract different components of the Mermaid code:

- **Node initialization statements** - Extracts node definitions and their types

- **Class definition statements** - Identifies styling classes for nodes

- **Node class assignments** - Maps nodes to their respective classes

- **Node connection statements** - Extracts source, target, and label information from connections

Our validation system performs several critical checks:

1. **W1** - Verifies the presence of required PROBLEM and RETURN interface nodes

2. **W2** - Ensures each node is properly connected in the workflow with paths from PROBLEM and to RETURN

3. **W3** - Validates that PROBLEM and RETURN nodes are correctly classified as Interface types

4. **W4** - Checks that all nodes have valid types according to configuration

5. **W5** - Ensures ScEnsembleOp nodes have at least two incoming connections

We also implement rigorous hard checks by compiling the Mermaid code directly through the Mermaid CLI[2], which thoroughly verifies syntactic correctness and catches any compilation errors. This comprehensive two-level validation approach ensures both structural integrity through soft checks and strict syntactic validity through hard checks, guaranteeing that all workflow diagrams are executable and error-free.

## A.3 Algorithms

In this section, we define the key algorithms underlying the MermaidFlow system. These algorithms form the core components of our workflow optimization and execution process, including the end-to-end MermaidFlow pipeline and the dedicated Mermaid workflow optimization routines that operate over Mermaid graphs. The following algorithms demonstrate how we generate, validate, and optimize workflow, as well as how we transform the Mermaid code into executable Python code. [3]

---

**Algorithm 1** MermaidFlow **End-to-End Pipeline**

---

1: **procedure** MERMAIDFLOWPIPELINE
2:     **Input:** val_dataset, dataset_type, mmd_checker, max_rounds, num_tries, candi_num
3:     initial_workflow ← **LoadInitialWorkflow**(dataset_type)
4:     mmd_history ← [initial_workflow]
5:     **for** round = 1 to max_rounds **do**
6:         parent_info ← **SampleParent**(mmd_history)
7:         mmd_graph, related_prompt ← **OptimizeMermaidWorkflow**(parent_info, num_tries, candi_num, mmd_checker)
8:         python_code ← **GeneratePythonCode**(parent_info, mmd_graph, related_prompt)
9:         val_score ← **EvaluateWorkflow**(python_code, validation_dataset)
10:        mmd_history ← **Update**(mmd_history, mmd_graph, python_code, val_score)
11:     **end for**
12:     **return** mmd_history
13: **end procedure**

---

[2]https://github.com/mermaid-js/mermaid-cli
[3]We use "mmd" as the abbreviation for Mermaid.

---

**Algorithm 2 OptimizeMermaidWorkflow** Function

    **Input:** parent_info, num_tries, candi_num, mermaid_checker
1: last_mmd, last_errors ← null
2: prev_attempt ← {last_mmd, last_errors}
3: **for** $i = 1$ to num_tries **do**
4:     mmd_graph, related_prompt ← **GetNewMermaidGraph**(parent_info, prev_attempt, candi_num)
5:     **if** mermaid_checker exists **then**
6:         mmd_path ← mermaid_checker.**TransferMmdCodeToFile**(mmd_graph)
7:         (hard_pass, hard_info) ← mermaid_checker.**HardCheck**(mmd_path)
8:         (soft_pass, soft_info) ← mermaid_checker.**SoftCheck**(mmd_path)
9:         last_mmd ← mmd_graph
10:       last_errors ← hard_info + soft_info
11:       prev_attempt ← {last_mmd, last_errors}
12:       **if** hard_pass AND soft_pass **then**
13:         **break**              ▷ The new Mermaid Code pass soft and hard check
14:       **end if**
15:     **else**
16:       **break**                   ▷ Accept first response if no checker
17:     **end if**
18: **end for**
19: **return** mmd_graph, related_prompt

---

---

**Algorithm 3 GetNewMermaidGraph** Function

1: **procedure** GETNEWMERMAIDGRAPH(parent_info, prev_attempt, candi_num)
2:     mmd_candidates ← **GetMermaidCandidates**(parent_info, prev_attempt, candi_num)
3:     selected_mmd ← **LLMAsJudge**(mmd_candidates)
4:     new_graph ← selected_mmd.$graph$
5:     new_prompt ← selected_mmd.$prompt$
6:     **return** new_graph, new_prompt
7: **end procedure**

---

### A.3.1 PROMPT TEMPLATES

We design structured prompt templates to guide the LLM in generating, evaluating, and transforming workflows throughout the optimization process. Each template encodes precise instructions, including output format, semantic constraints, and evaluation objectives, ensuring that the model adheres to the structural and functional requirements of MermaidFlow.

Below, we summarize the purpose of each template and provide the full prompt text for reference.

- **UPDATE_MERMAID_WORKFLOW** - Used to guide the LLM in optimizing mermaid code based on parent nodes information. This prompt includes experience data, score information, the current graph in Mermaid format, the corresponding role prompt, operator descriptions, and log data from previous runs. It instructs the LLM to make specific optimizations by adding, modifying, or deleting nodes while clearly describing each modification. The prompt also emphasizes the importance of proper formatting, critical thinking methods, and effective use of specialized operators like Program for solving math problems.

- **MERMAID_GUIDANCE** - Used to guide the generation of correct Mermaid code according to our node type definitions. This prompt provides specific guidance for generating workflow in Mermaid code following the specification in A.1.

- **GENERATE_PYTHON_CODE** - Used to generate python code from given mermaid code and agents prompt.

- **LLM_AS_JUDGE** - Used to select the most promising graph from multiple Mermaid candidates. It evaluates each candidate graph based on criteria including workflow coherence,

innovation, complexity balance, prompt quality, and modification rationale, providing detailed scoring justifications. The judging process references the performance of historical graphs and avoids selecting graphs with structural defects (such as incorrect connections or improper use of ensemble nodes).

**UPDATE_MERMAID_WORKFLOW** : Depending on whether a single or two parent workflows are used, we adopt different prompt templates to generate the new Mermaid code.

**Single Parent**

```
UPDATE_MERMAID_WORKFLOW_MATH = """
You are building a Graph and corresponding Prompt to jointly solve {type} problems.
↪  Here is a graph written in Mermaid and the corresponding prompt that performed
↪  excellently in a previous iteration (maximum score is 1). There are also some
↪  special operators defined in the operator_description section. Referring to this
↪  given graph and prompt, which forms a basic example of a code solution approach,
↪  please reconstruct and optimize them.
You should make further optimizations based on this graph. The modified graph should
↪  differ from the provided example or have the same architecture but with prompts
↪  fine-tuned based on logs. The specific differences should be noted within the
↪  <modification>xxx</modification> section.
<sample>
    <experience>{experience}</experience>
    <modification>(such as: add/delete/modify/...)</modification>
    <score>{score}</score>
    <graph>{graph_mermaid}</graph>
    <role_prompt>{prompt}</role_prompt>(only prompt_custom)
    <operator_description>{operator_description}</operator_description>
</sample>
Below are the logs of some results with the aforementioned Graph that performed well
↪  but encountered errors, which can be used as references for optimization:
{log}

You can add, modify, or delete nodes, parameters, or connections in the workflow. For
↪  each change, clearly describe your single modification within XML tags using the
↪  format: <modification>your detailed modification description</modification>.

Your optimizations should be one of (this limitation is strict! you can only pick one
↪  action!):
1. Expanding the graph by adding new single (operators). Note: adding nodes may require
↪  modifying prompts for related nodes, you should also update the corresponding
↪  prompts if needed.
2. Deleting unnecessary nodes from the graph.
3. Modifying existing nodes or their connections or their prompt.

Prompt is very important for performance here are some exmaples you can learn from it:
{prompt_few_shot}

Ensure all necessary prompts are properly prepared and defined. The generated custom
↪  node's role is defined by their prompt. Use custom methods with proper formatting
↪  to ensure your output matches the expected structure. The system will extract
↪  answers based on specific rules for scoring, so maintaining the correct output
↪  format is critical.
The output format is critical for proper evaluation. Analyze the log data carefully to
↪  identify patterns in successful solutions and common errors. Extract specific
↪  formatting requirements and incorporate them as clear guidance in your prompts.
When optimizing, you can incorporate critical thinking methods like review, revise,
↪  ensemble (generating multiple answers through different/similar prompts, then
↪  voting/integrating/checking the majority to obtain a final answer), selfAsk, etc.

NOTE: You should also try to use different operators to enhance workflow capabilities.
NOTE: If you are trying to answer from multiple solutions, you should try to use
↪  ensemble operator then to get the final answer.
NOTE: Each output of nodes except the interface node should be connected to the next
↪  node, ensuring data flows through the entire workflow. This guarantees all nodes
↪  properly receive inputs and pass outputs.
NOTE: If you are trying to add an ensemble node, you need to guarantee that there are
↪  multiple solution inputs available for the ensemble to work effectively.
NOTE: Program is a very powerful tool for solving MATH problems, you should try it! The
↪  Program operator can execute Python code to perform calculations and solve complex
↪  mathematical problems.
NOTE: Think big! Be bold in innovation and exploration. Don't be afraid to push
↪  boundaries and discover new approaches to problem-solving. Additionally, keep
↪  prompts concise - no more than 100 words per prompt. Shorter, focused prompts often
↪  perform better than lengthy ones.
"""
```

```
UPDATE_MERMAID_WORKFLOW_CODE = """
You are building a Graph and corresponding Prompt to jointly solve {type} problems.
↪  Here is a graph written in Mermaid and the corresponding prompt that performed
↪  excellently in a previous iteration (maximum score is 1). There are also some
↪  special operators defined in the operator_description section. Referring to this
↪  given graph and prompt, which forms a basic example of a code solution approach,
↪  please reconstruct and optimize them.
You should make further optimizations based on this graph. The modified graph should
↪  differ from the provided example or have the same architecture but with prompts
↪  fine-tuned based on logs. The specific differences should be noted within the
↪  <modification>xxx</modification> section.
<sample>
    <experience>{experience}</experience>
    <modification>(such as: add/delete/modify/...)</modification>
    <score>{score}</score>
    <graph>{graph_mermaid}</graph>
    <role_prompt>{prompt}</role_prompt>(only prompt_custom)
    <operator_description>{operator_description}</operator_description>
</sample>
Below are the logs of some results with the aforementioned Graph that performed well
↪  but encountered errors, which can be used as references for optimization:
{log}

You can add, modify, or delete nodes, parameters, or connections in the workflow. For
↪  each change, clearly describe your single modification within XML tags using the
↪  format: <modification>your detailed modification description</modification>.

Your optimizations should be one of (this limitation is strict! you can only pick one
↪  action!):
1. Expanding the graph by adding new single (operators). Note: adding nodes may require
↪  modifying prompts for related nodes, you should also update the corresponding
↪  prompts if needed.
2. Deleting unnecessary nodes from the graph.
3. Modifying existing nodes or their connections or their prompt.

Prompt is very important for performance here are some exmaples you can learn from it:
{prompt_few_shot}

Ensure all necessary prompts are properly prepared and defined. The generated custom
↪  node's role is defined by their prompt. Use custom methods with proper formatting
↪  to ensure your output matches the expected structure. The system will extract
↪  answers based on specific rules for scoring, so maintaining the correct output
↪  format is critical.
Review the log data carefully to understand the expected answer format and ensure your
↪  implementation produces compatible output. Proper formatting is essential for
↪  accurate evaluation of your solution, and you should emphasize this in the prompt.
When optimizing, you can incorporate critical thinking methods like review, revise,
↪  ensemble (generating multiple answers through different/similar prompts, then
↪  voting/integrating/checking the majority to obtain a final answer), selfAsk, etc.
You should also try to use different operators to enhance workflow capabilities. Each
↪  operator should have the corresponding Mermaid class defined in the graph code.

NOTE: If you are trying to add an ensemble node, you need to guarantee that there are
↪  multiple solution inputs available for the ensemble to work effectively.
NOTE: In the code problems, ensemble different kinds of solutions is a good idea, and
↪  you should also try to use Test operator to validate the solutions.
NOTE: For code-related tasks, the final output must be either test_result["solution"]
↪  or custom_code_generate_result["response"], as these represent valid Python code.
↪  Make sure your workflow produces one of these formats as the final output to ensure
↪  proper evaluation.
NOTE: Think big! Be bold in innovation and exploration. Don't be afraid to push
↪  boundaries and discover new approaches to problem-solving. Additionally, keep
↪  prompts concise - no more than 100 words per prompt. Shorter, focused prompts often
↪  perform better than lengthy ones.
"""
```

**Two Parents**

```
UPDATE_MERMAID_WORKFLOW = """
# Evolution-Based Graph Generation

You are tasked with generating a new workflow graph by combining elements from two
↪  parent graphs. This process is inspired by genetic algorithms where offspring
↪  inherit traits from both parents.
This evolutionary approach allows us to create new solutions that may be more effective
↪  than either parent graph.
```

```
<parent_graph>
    <graph_A>
        <graph>{graph_A}</graph>
        <prompt>{prompt_A}</prompt>
        <score>{score_A}</score>
    </graph_A>
    <graph_B>
        <graph>{graph_B}</graph>
        <prompt>{prompt_B}</prompt>
        <score>{score_B}</score>
    </graph_B>
</parent_graph>

## Your Task
1. Analyze both **parent graphs** carefully (structure, nodes, connections, prompts,
↪  and purpose)
2. Create a new graph that:
    - Combines strengths from both parents
    - Introduces strategic innovations for improved performance
    - Applies evolutionary operations such as:
      * Crossover: Merging effective sections(nodes) from both parents, you should
      ↪  separate components in a reasonable and logical way, typically at interface or
      ↪  ensemble nodes
      * Mutation: Making targeted modifications
      * Insertion: Adding beneficial new nodes
      * Deletion: Removing inefficient components

## Guidelines
- Focus on the strengths of both parent graphs, especially the higher-scoring parent,
↪  to inform your design decisions
- Ensure your new graph is structurally correct and follows proper workflow patterns
- If creating a significantly different structure proves challenging, you may maintain
↪  a similar structure to the parents but with improved prompts, or, you can simply
↪  add one extra node to ensemble
- Pay careful attention to the connections between nodes to ensure proper data flow -
↪  for example, ensemble nodes should have multiple inputs
- Verify that your graph has correct input/output relationships and follows the
↪  operator usage guidelines
- Use custom methods to restrict your output format, rather than using code (outside of
↪  the code, the system will extract answers based on certain rules and score them)

## Prompt Guidence
- **Prompt engineering is crucial for performance**: Carefully analyze and learn from
↪  the prompts used in **high-scoring** historical workflows. Pay special attention to
↪  their structure, specificity, and instruction clarity. Here are some exemplary
↪  prompts you can use as reference:
    {prompt_few_shot}

Your response should include:

1. A detailed explanation of your modifications in the <modification> section
2. The complete Mermaid code for the new graph
3. The updated prompts for any custom nodes

NOTE: Ensure the new graph is valid Mermaid syntax and represents a complete workflow
↪  solution. The following section contains critical rules and guidance for using
↪  operators in Mermaid that you MUST follow to create an effective workflow:
{mermiad_custom_prompt}
"""
```

## MERMAID_GUIDANCE

**Mermaid Code Guidance**

```
MERMAID_CUSTOM_MATH = """
# Mermaid Graph Style Guide for MATH Problems
# This comprehensive guide defines styling and structure for creating consistent
↪  workflow diagrams

# Node Style Definitions
classDef CustomOp fill:#d0e1f9,stroke:#4378a2,stroke-width:2px;
classDef ProgrammerOp fill:#f9c2c2,stroke:#c23737,stroke-width:2px;
classDef ScEnSembleOp fill:#f9e4b7,stroke:#b99b37,stroke-width:2px;
classDef Interface fill:#e2e2f2,stroke:#6a6ab2,stroke-width:2px;

# ===== OPERATOR USAGE GUIDE =====

# 1. Interface Nodes (Entry/Exit Points)
```

```
972
973    # Every workflow diagram must include these two standard interface nodes:
       #   PROBLEM([Problem])              - The entry point providing initial input
974    #   RETURN([Return response & cost]) - The exit point receiving final output
       #
975    # Example:
976    #   PROBLEM([Problem])
       #   RETURN([Return response & cost])
977    #
978    #   class PROBLEM Interface
       #   class RETURN Interface
979    #
980    # Connection rules:
       #   - PROBLEM node: Provides input to other nodes but never receives input
981    #   - RETURN node: Receives output from the final node but never produces output
982
       # 2. Custom Operator Nodes
983    # Format: NodeName["Custom
(role: role_name)"]
984    #
       # Example:
985    #   K["Custom
(role: validate_1)"]
986    #   class K CustomOp
       #
987    # For multiple nodes with similar roles, use numbered suffixes:
988    #   R1["Custom
(role: review_solution_1)"]
       #   R2["Custom
(role: review_solution_2)"]
989    #   class R1 CustomOp
990    #   class R2 CustomOp
       #
991    # Connection example:
992    #   PROBLEM --> |input| R1
993    # 3. Programmer Operator Nodes
994    # Format: P["Programmer
(analysis: 'your analysis text')"]
       #
995    # The Programmer operator requires two inputs:
996    # - problem: The math problem to solve
       # - analysis: Instructions on how to approach the problem
997    #
998    # Examples:
       #   P["Programmer
(analysis: 'Calculate step by step')"]
999    #   class P ProgrammerOp
1000   #
       # Connection rules:
1001   # 1. For the problem input:
1002   #   PROBLEM --> |problem|P  # The problem must come from the PROBLEM node
       #
1003   # 2. For the analysis input (two options):
       #   C --> |analysis|P  # You can use another node's output as analysis content
1004   #   # OR
1005   #   # You can specify the analysis directly in the node definition
       #
1006   # Complete example:
       #   PROBLEM --> |problem|P
1007   #   C --> |analysis|P
1008   #   class P ProgrammerOp
1009   # 4. ScEnsemble Operator Nodes
1010   # Format: ENSEMBLE["ScEnsemble
"]
       #
1011   # Example:
       #   ENSEMBLE["ScEnsemble
"]
1012   #   class ENSEMBLE ScEnSembleOp
1013   #
       # CRITICAL: ScEnsemble nodes MUST have multiple inputs to function correctly
1014   # Example connections:
1015   #   SOLUTION_1 --> ENSEMBLE
       #   SOLUTION_2 --> ENSEMBLE
1016   #   ENSEMBLE --> NEXT_NODE
1017
       # ===== WORKFLOW PATTERNS =====
1018
1019   # Example Workflow Pattern
       # This pattern demonstrates how to effectively combine multiple solution approaches:
1020   # 1. Problem input flows to multiple solution-generating nodes (Custom and/or
       ↪  Programmer)
1021
       # 2. All solutions are combined using ScEnsemble
1022   # 3. The ensemble result is returned as the final output
       #
1023   # Connection pattern:
1024   #   PROBLEM --> |input| SOLUTION_1
       #   PROBLEM --> |input| SOLUTION_2
1025
```

```
1026
1027        #    PROBLEM --> |problem| P
1028        #    SOLUTION_1 --> ENSEMBLE
           #    SOLUTION_2 --> ENSEMBLE
1029        #    P --> ENSEMBLE
           #    ENSEMBLE --> RETURN
1030
1031        # ===== GENERAL RULES =====

1032        # Connection Rules
           # 1. Direct connections from PROBLEM to any node that needs the initial problem as
1033        ↪  input
1034        # 2. Direct connections between nodes where one node requires another's output
           # 3. When using ProgrammerOp, the edges should have name on it
1035        # 4. All connections must follow logical data flow and maintain workflow coherence

1036
           # Prompt Definition Requirements
1037        # All prompts used in the graph must be defined in this format:
           # <prompt>
1038        # PROMPT_NAME_1="Your first prompt text here"
1039        # PROMPT_NAME_2="Your second prompt text here"
           # </prompt>
1040        #
1041        # Each prompt must be on a separate line with its own unique variable name

1042        # IMPORTANT: Do not create new Operator types!
1043        # Only use the predefined operators available in the system. Creating custom operators
           # will cause workflow execution failures. Stick to the operators documented in this
1044        ↪  guide.

1045
           # Best Practices:
1046        # - Use %% for comments in separate lines for clarity, don't add it to the end of the
1047        ↪  line because it is not a valid mermaid code
           # - Always include the style block (classDef section) even if not all classes are used
1048        # - Always ensure multiple inputs to ScEnsemble nodes (this is a common mistake)
           # - Maintain consistent naming conventions for all nodes and classes
1049        # - Use descriptive role names that indicate the node's purpose
           # - Label connections with |input| where appropriate for clarity
1050

1051

1052        """
           MERMAID_CUSTOM_CODE = """
1053        # Operator Style Definitions
           # Each operator has a specific style for consistent visualization in the Mermaid graph
1054        classDef CustomOp fill:#d0e1f9,stroke:#4378a2,stroke-width:2px;
           classDef CustomCodeGenerateOp fill:#f9c2c2,stroke:#c23737,stroke-width:2px;
1055        classDef ScEnSembleOp fill:#f9e4b7,stroke:#b99b37,stroke-width:2px;
1056        classDef TestOp fill:#d8f0d8,stroke:#2e8b57,stroke-width:2px;
           classDef Interface fill:#e2e2f2,stroke:#6a6ab2,stroke-width:2px;
1057

1058        # ===== OPERATOR USAGE GUIDE =====
1059
           # 1. Interface Nodes
1060        # Interface nodes represent the entry and exit points of your workflow
1061        # Required in every graph:
           #    PROBLEM([Problem])          - Starting point that provides input
1062        #    RETURN([Return response & cost]) - Endpoint that receives final output
           #
1063        # Example:
1064        #    class PROBLEM Interface
           #    class RETURN Interface
1065        #
1066        # Connection rules:
           #    - PROBLEM node provides input but doesn't receive any
1067        #    - RETURN node receives output but doesn't produce any
1068
           # 2. Custom Operator
1069        # Used for specialized reasoning strategies with defined roles
1070        #
           # Example:
1071        #    K["Custom
(role: xxx)"]
           #    class K CustomOp
1072        #
1073        # For multiple instances with same prompt type:
           #    R1["Custom
(role: xxx_1)"]
1074        #    R2["Custom
(role: xxx_2)"]
1075        #    class R1 CustomOp
           #    class R2 CustomOp
1076        #
1077        # Connection rules:
1078        #    - Connect directly to PROBLEM if it needs the initial problem
           #    - Connect to other nodes if it needs their output
1079
```

```
# 3. CustomCodeGenerate Operator
# Specialized for code generation tasks based on problem descriptions
#
# Example:
#   CODE_GEN["CustomCodeGenerate
(instruction: xxx)"]
#   class CODE_GEN CustomCodeGenerateOp
#
# Multiple approaches:
#   CODE_GEN_1["CustomCodeGenerate
(instruction: aaa)"]
#   CODE_GEN_2["CustomCodeGenerate
(instruction: bbb)"]
#   class CODE_GEN_1 CustomCodeGenerateOp
#   class CODE_GEN_2 CustomCodeGenerateOp
#

# 4. ScEnsemble Operator
# Combines multiple solutions into one cohesive result
#
# Example:
#   ENSEMBLE["ScEnsemble
"]
#   class ENSEMBLE ScEnsembleOp
#
# CRITICAL: Must have multiple inputs to function correctly
#   SOLUTION_1 --> ENSEMBLE
#   SOLUTION_2 --> ENSEMBLE
#   ENSEMBLE --> other_node

# 5. Test Operator
# Validates solutions against test cases
#
# Example:
#   T["Test
"]
#   class T TestOp
#
# Typical connection pattern:
#   CODE_GEN --> |solution|T
#   ENTRY_POINT --> |entry_point|T
#   PROBLEM --> |problem|T
#   T --> DECISION_NODE
#
# Decision node for test results:
#   CHECK_TEST{test_result
passed?}
#   class CHECK_TEST DecisionOp
#   CHECK_TEST -- Failed --> IMPROVE_SOLUTION
#   CHECK_TEST -- Passed --> RETURN

# ===== IMPORTANT NOTES =====
# - You cannot create other class operations
# - Always ensure multiple inputs to ScEnsemble nodes
# - All prompts must be defined in the prompt section
# - Format prompts as:
#   <prompt>
#   PROMPT_NAME_1="Your first prompt text here"
#   PROMPT_NAME_2="Your second prompt text here"
#   </prompt>

# IMPORTANT: Do not create new Operator types!
# Only use the predefined operators available in the system. Creating custom operators
# will cause workflow execution failures. Stick to the operators documented in this
↪  guide.
"""
```

## GENERATE_PYTHON_CODE

**Generate Python Code**

```
GENERATE_PYTHON_CODE = """
Below is a graph, the corresponding Python code, and the prompt.
<old_workflow>
    <old_graph>{old_mermaid}</old_graph>
    <old_code>{old_code}</old_code>
    <old_role_prompt>{old_prompt}</old_role_prompt>(only prompt_custom)
</old_workflow>

Based on this example of old graph and code, you need to generate new Python code
↪  according to the new graph and given prompt.

<information_for_new_workflow>
    <new_graph>{new_mermaid}</new_graph>
```

```
      <new_role_prompt>{new_prompt}</new_role_prompt>(only prompt_custom)
      <operator_description>{operator_description}</operator_description>
</information_for_new_workflow>

The output format should be:

New generated python code

Carefully analyze the new_graph to generate corresponding code. Pay attention to:
1. The connections between each node
2. The role and function of each operator
3. The input/output relationships

For each node in the graph, implement the appropriate code and use the corresponding
↪   prompts from new_prompt. If a node or operator doesn't have an explicit prompt
↪   provided, DO NOT create one - use empty strings instead. For example, Program
↪   operators may have empty analysis fields.
Every prompt referenced in your Python code must be defined in the prompt_custom
↪   module. When adding new functionality to your Python code, make sure to import any
↪   necessary libraries or modules, except for operator, prompt_custom,
↪   create_llm_instance, and CostManage which are automatically imported.
Your implementation must be robust - ensure all methods return appropriate values and
↪   **never return None for any field**. Pay special attention to error handling and
↪   edge cases.
Use custom methods with proper formatting to ensure your output matches the expected
↪   structure. The system will extract answers based on specific rules for scoring, so
↪   maintaining the correct output format is critical.

NOTE: especially for the final output, you should ensure that the output format is
↪   correct, you can learn it from old code.
"""
```

## LLM_AS_JUDGE

**LLM As Judge**

```
LLM_AS_JUDGER = """

Your task is to select the most promising graph from the candidates. Here is the
↪   content of each graph, written in Mermaid code, along with explanations of how each
↪   new graph was generated from parent graphs:
Each new graph is derived from two parent graphs, with parent graph information as
↪   follows:

<parent_graph>
      <parent_graph_A>{parent_graph_A}</parent_graph_A>
      <parent_prompt_A>{parent_prompt_A}</parent_prompt_A>

      <parent_graph_B>{parent_graph_B}</parent_graph_B>
      <parent_prompt_B>{parent_prompt_B}</parent_prompt_B>
</parent_graph>

<graph_candidates>
      <graph_A>{graph_A}</graph_A>
      <modification_A>{modification_A}</modification_A>
      <prompt_A>{prompt_A}</prompt_A>

      <graph_B>{graph_B}</graph_B>
      <modification_B>{modification_B}</modification_B>
      <prompt_B>{prompt_B}</prompt_B>

      <graph_C>{graph_C}</graph_C>
      <modification_C>{modification_C}</modification_C>
      <prompt_C>{prompt_C}</prompt_C>

      <graph_D>{graph_D}</graph_D>
      <modification_D>{modification_D}</modification_D>
      <prompt_D>{prompt_D}</prompt_D>
</graph_candidates>

Please evaluate the graph candidates and select the most promising one based on these
↪   criteria:

1. **Workflow Coherence**: Assess how well the nodes connect and form a logical
↪   workflow
```

```
2. **Innovation**: Evaluate how the new graph improves upon the parent graphs
3. **Complexity Balance**: Check if the graph has appropriate complexity (neither too
↪  simple nor unnecessarily complex)
4. **Prompt Quality**: Examine the quality and specificity of the node prompts
5. **Modification Rationale**: Consider the thoughtfulness of the explanation provided
↪  for the changes

Here are some history of previous graphs and their corresponding score:
<history>
    {elites_history}
</history>

For each candidate graph, provide a score from 1-10 for each criterion and explain your
↪  reasoning, you can learn from the history graphs.
You should avoid selecting graphs with structural defects, such as:
1. Incorrectly connecting nodes (e.g., CustomOp should not directly feed into
↪  ProgrammerOp)
2. Not properly using the ensemble node (all solution-generating nodes should feed into
↪  it)
3. Missing critical connections between nodes
4. Creating circular dependencies

Here are the specific structural rules for this type of workflow:
{mermaid_usage}

Additionally, consider how well the graph follows established patterns from successful
↪  historical examples.

Then select the graph with the highest total score as the most promising candidate.

<evaluation>
    <graph_A_score>
        <workflow_coherence>Score (1-10)</workflow_coherence>
        <innovation>Score (1-10)</innovation>
        <complexity_balance>Score (1-10)</complexity_balance>
        <prompt_quality>Score (1-10)</prompt_quality>
        <modification_rationale>Score (1-10)</modification_rationale>
        <total_score>Sum of all scores</total_score>
        <explanation>Detailed explanation of your evaluation</explanation>
    </graph_A_score>

    <graph_B_score>
        <workflow_coherence>Score (1-10)</workflow_coherence>
        <innovation>Score (1-10)</innovation>
        <complexity_balance>Score (1-10)</complexity_balance>
        <prompt_quality>Score (1-10)</prompt_quality>
        <modification_rationale>Score (1-10)</modification_rationale>
        <total_score>Sum of all scores</total_score>
        <explanation>Detailed explanation of your evaluation</explanation>
    </graph_B_score>

    <graph_C_score>
        <workflow_coherence>Score (1-10)</workflow_coherence>
        <innovation>Score (1-10)</innovation>
        <complexity_balance>Score (1-10)</complexity_balance>
        <prompt_quality>Score (1-10)</prompt_quality>
        <modification_rationale>Score (1-10)</modification_rationale>
        <total_score>Sum of all scores</total_score>
        <explanation>Detailed explanation of your evaluation</explanation>
    </graph_C_score>

    <graph_D_score>
        <workflow_coherence>Score (1-10)</workflow_coherence>
        <innovation>Score (1-10)</innovation>
        <complexity_balance>Score (1-10)</complexity_balance>
        <prompt_quality>Score (1-10)</prompt_quality>
        <modification_rationale>Score (1-10)</modification_rationale>
        <total_score>Sum of all scores</total_score>
        <explanation>Detailed explanation of your evaluation</explanation>
    </graph_D_score>
</evaluation>

<selected_graph>[A/B/C/D]</selected_graph>
<justification>
    Please provide a comprehensive justification for your selection, highlighting the
    ↪  key strengths of the chosen graph and how it represents the most effective
    ↪  approach to solving the problem.
</justification>
"""
```

### A.4    DETAILS OF BASELINE METHODS

In this section, we provide a detailed description of the configurations for baseline methods:

- **CoT** (Kojima et al., 2022). Chain-of-Thought prompting guides the LLM to decompose complex reasoning into a sequence of intermediate steps, rather than generating a direct answer.
- **ComplexCoT** (Fu et al., 2023). This method builds on CoT by explicitly modeling and controlling reasoning complexity across multiple steps. We use the official implementation from `https://github.com/FranxYao/Complexity-Based-Prompting/tree/main`.
- **Self-Consistency** (Wang et al., 2023). This method generates five independent chain-of-thought trajectories by sampling the LLM multiple times, then consolidates them via majority voting to improve the reliability of the final answer.
- **LLM-Debate** (Du et al., 2024). A panel of five role-specific LLM agents engage in up to two rounds of structured debate; the final answer is selected by majority vote.
- **DyLAN** (Liu et al., 2024). DyLAN employs a dynamic multi-agent framework in which agents iteratively share and refine intermediate reasoning to converge on high-quality solutions.
- **MacNet** (Qian et al., 2024). MacNet structures its agent network as a fully meshed graph, enabling every agent to exchange and integrate information with all others at each reasoning step.
- **GPTSwarm** (Zhuge et al., 2024). GPTSwarm leverages a swarm-intelligence paradigm by orchestrating multiple LLM agents in parallel; agents independently propose solutions and iteratively refine them through shared feedback, following the protocol outlined in the original paper.
- **MaAS** (Zhang et al., 2025b). MaAS integrates a trainable module for dynamic workflow assignment across multiple agents; we replicate the authors' original architecture using their official code and retain their specified hyperparameters without modification.
- **AutoAgents** (Chen et al., 2024). AutoAgents orchestrates a pipeline of specialized LLM agents that autonomously schedule, execute, and refine subtasks through coordinated interactions; we replicate the authors' default configuration and parameter settings from the official repository.
- **ADAS** (Hu et al., 2024). ADAS employs an adaptive debate-and-selection mechanism among multiple LLM agents, dynamically refining candidate solutions based on quality criteria.
- **AFlow** (Zhang et al., 2024c). In the original AFlow study, both gpt-4o-mini and claude-3.5-sonnet are used. For a fair comparison, we constrain AFlow to gpt-4o-mini only and set MAX_ITERATION to 20.

### A.5    DETAILS OF TASKS AND BENCHMARKS

Following established methodologies in workflow automation (Zhang et al., 2024c; 2025b), we partition each dataset into training and test sets with a TRAIN:TEST ratio of 1:4. For the MATH benchmark, we follow the approach in (Zhang et al., 2025b), selecting a subset of 617 challenging problems across four representative categories: Combinatorics & Probability, Number Theory, Pre-algebra, and Pre-calculus, all at difficulty level 5. The dataset statistics are summarized in Table 4.

## B    CASE STUDY

In this section, we present examples of the optimal workflows discovered by our approach for each of the four datasets. For each dataset, we provide the **Mermaid code**, corresponding **Python code**, and the rendered **workflow diagram**.

The declarative Mermaid representation serves as the backbone of structured workflow generation, enabling several crucial capabilities:

Table 4: Statistics of datasets used in our experiments.

| Domain | Dataset | Training | Testing | Evaluation Metric |
|---|---|---|---|---|
| Code Generation | HumanEval | 33 | 131 | pass@1 |
| | MBPP | 86 | 341 | pass@1 |
| Math Reasoning | GSM8K | 264 | 1,055 | Accuracy |
| | MATH | 119 | 486 | Accuracy |

1. **Typed, Semantically Aligned Representation.** Mermaid encodes workflows as statically typed and semantically annotated graphs, allowing prompt semantics and operator roles to be aligned explicitly within the workflow structure.

2. **Human-Readable and Verifiable Syntax.** Unlike imperative representations, Mermaid provides a format that is both visually interpretable and statically verifiable, facilitating intuitive debugging, planning, and workflow validation.

3. **Reliable Code Translation.** The structured nature of Mermaid graphs enables seamless compilation into executable Python code, with consistent type and role guarantees that reduce runtime errors and improve reliability.

These properties make MermaidFlow not just a search mechanism, but a principled framework for building modular, interpretable, and safer agentic workflows. The case studies illustrate how these benefits translate into practical gains across crucial domains such as programming, math, and symbolic reasoning.

## B.1 GSM8K

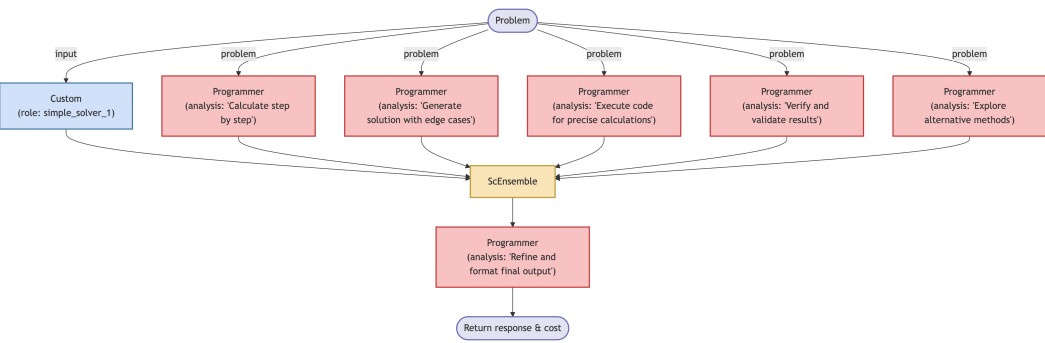

Figure 5: Mermaid diagram for GSM8K.

**Mermaid Code**

```
flowchart TD
%% Nodes
PROBLEM([Problem])
C["Custom
(role: simple_solver_1)"]
```

```
P1["Programmer
(analysis: 'Calculate step by step')"]
P2["Programmer
(analysis: 'Generate solution with edge cases')"]
P3["Programmer
(analysis: 'Execute code for precise calculations')"]
P4["Programmer
(analysis: 'Verify and validate results')"]
P5["Programmer
(analysis: 'Explore alternative methods')"]
P6["Programmer
(analysis: 'Refine and format final output')"]
ENSEMBLE["ScEnsemble
"]
RETURN([Return response & cost])

%% Styles
classDef CustomOp fill:#d0e1f9,stroke:#4378a2,stroke-width:2px;
classDef ProgrammerOp fill:#f9c2c2,stroke:#c23737,stroke-width:2px;
classDef ScEnSembleOp fill:#f9e4b7,stroke:#b99b37,stroke-width:2px;
classDef Interface fill:#e2e2f2,stroke:#6a6ab2,stroke-width:2px;

%% Assign classes
class C CustomOp
class P1 ProgrammerOp
class P2 ProgrammerOp
class P3 ProgrammerOp
class P4 ProgrammerOp
class P5 ProgrammerOp
class P6 ProgrammerOp
class ENSEMBLE ScEnSembleOp
class PROBLEM Interface
class RETURN Interface

%% Flow (arrows show data relationships)
PROBLEM --> |input|C
PROBLEM --> |problem|P1
PROBLEM --> |problem|P2
PROBLEM --> |problem|P3
PROBLEM --> |problem|P4
PROBLEM --> |problem|P5
C --> ENSEMBLE
P1 --> ENSEMBLE
P2 --> ENSEMBLE
P3 --> ENSEMBLE
P4 --> ENSEMBLE
P5 --> ENSEMBLE
ENSEMBLE --> P6
P6 --> RETURN
```

**Python Code**

```python
from typing import Literal
import workspace.GSM8K.workflows.template.operator as operator
import workspace.GSM8K.workflows.round_16.prompt as prompt_custom
from scripts.async_llm import create_llm_instance
import weave

DatasetType = Literal["HumanEval", "MBPP", "GSM8K", "MATH", "HotpotQA", "DROP"]

class Workflow:
    def __init__(
        self,
        name: str,
        llm_config,
        dataset: DatasetType,
    ) -> None:
        self.name = name
        self.dataset = dataset
        self.llm = create_llm_instance(llm_config)
        self.sc_ensemble = operator.ScEnsemble(self.llm)
        self.custom = operator.Custom(self.llm)
        self.programmer = operator.Programmer(self.llm)

    @weave.op()
    async def __call__(self, problem: str):
        """
        Implementation of the workflow
        Each operator is callable, you can call it directly.
        """
        # Step 1: Use the Custom operator to generate a detailed solution
        custom_response = await self.custom(input=problem,
        ↪   instruction=prompt_custom.SIMPLE_SOLVER_1, role="simple_solver_1")
```

```
        # Step 2: Use the Programmer operator to analyze the problem and provide a code
        ↪   solution
        programmer_response_1 = await self.programmer(problem=problem,
        ↪   analysis="Calculate step by step")

        # Step 3: Use the Programmer operator to generate a solution considering edge
        ↪   cases
        programmer_response_2 = await self.programmer(problem=problem,
        ↪   analysis="Generate solution with edge cases")

        # Step 4: Use the Programmer operator to execute code for precise calculations
        programmer_response_3 = await self.programmer(problem=problem,
        ↪   analysis="Execute code for precise calculations")

        # Step 5: Use the Programmer operator to verify and validate results
        programmer_response_4 = await self.programmer(problem=problem, analysis="Verify
        ↪   and validate results")

        # Step 6: Use the Programmer operator to explore alternative methods
        programmer_response_5 = await self.programmer(problem=problem,
        ↪   analysis="Explore alternative methods")

        # Step 7: Combine the responses from Custom and all Programmer responses for
        ↪   the ScEnsemble
        solutions = [
            custom_response['response'],
            programmer_response_1['output'],
            programmer_response_2['output'],
            programmer_response_3['output'],
            programmer_response_4['output'],
            programmer_response_5['output']
        ]

        # Step 8: Use the ScEnsemble operator to select the best solution
        ensemble_response = await self.sc_ensemble(solutions=solutions,
        ↪   problem=problem)

        # Step 9: Refine and format the final output
        final_output = await self.programmer(problem=ensemble_response['response'],
        ↪   analysis="Refine and format final output")

        return final_output['output'], self.llm.get_usage_summary()["total_cost"]
```

**Corresponding Prompt**

```
SIMPLE_SOLVER_1 = """Please solve the given mathematical problem step by step. Follow
↪   these guidelines:

1. State the problem clearly.
2. Outline the approach and any relevant formulas or concepts.
3. Provide detailed calculations, using LaTeX notation for mathematical expressions.
4. Explain each step of your reasoning.
5. Verify and validate your results to ensure accuracy.
6. Present the final answer enclosed in \\boxed{} LaTeX notation.
7. Ensure all mathematical notation is in LaTeX format.

Your solution should be thorough, mathematically sound, and easy to understand."""
```

## B.2 MATH

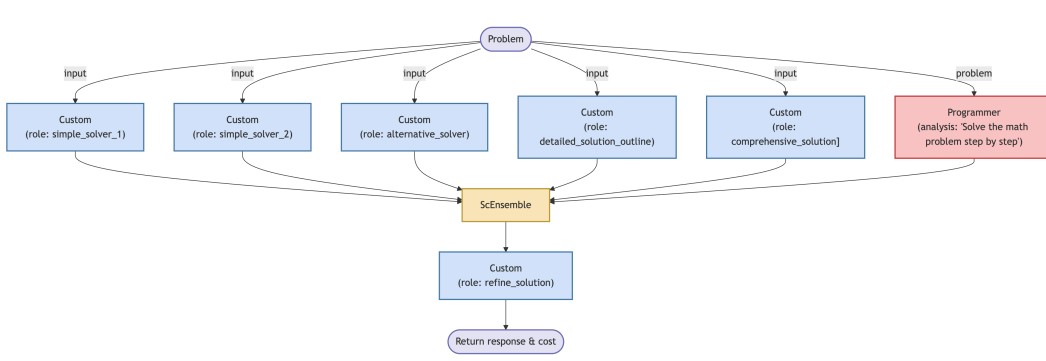

Figure 6: Mermaid diagram for MATH.

**Mermaid Code**

```
flowchart TD
    %% Nodes
    PROBLEM([Problem])
    C1["Custom
(role: simple_solver_1)"]
    C2["Custom
(role: simple_solver_2)"]
    C3["Custom
(role: alternative_solver)"]
    C4["Custom
(role: detailed_solution_outline)"]
    C5["Custom
(role: comprehensive_solution)"]
    P["Programmer
(analysis: 'Solve the math problem step by step')"]
    REFINE["Custom
(role: refine_solution)"]
    ENSEMBLE["ScEnsemble
"]
    RETURN([Return response & cost])

    %% Styles
    classDef CustomOp fill:#d0e1f9,stroke:#4378a2,stroke-width:2px;
    classDef ProgrammerOp fill:#f9c2c2,stroke:#c23737,stroke-width:2px;
    classDef ScEnSembleOp fill:#f9e4b7,stroke:#b99b37,stroke-width:2px;
    classDef Interface fill:#e2e2f2,stroke:#6a6ab2,stroke-width:2px;

    %% Assign classes
    class C1 CustomOp
    class C2 CustomOp
    class C3 CustomOp
    class C4 CustomOp
    class C5 CustomOp
    class P ProgrammerOp
    class REFINE CustomOp
    class PROBLEM Interface
    class RETURN Interface
    class ENSEMBLE ScEnSembleOp

    %% Flow (arrows show data relationships)
    PROBLEM --> |input|C1
    PROBLEM --> |input|C2
    PROBLEM --> |input|C3
    PROBLEM --> |input|C4
    PROBLEM --> |input|C5
    PROBLEM --> |problem|P
    C1 --> ENSEMBLE
    C2 --> ENSEMBLE
    C3 --> ENSEMBLE
```

```
        C4 --> ENSEMBLE
        C5 --> ENSEMBLE
        P --> ENSEMBLE
        ENSEMBLE --> REFINE
        REFINE --> RETURN
```

**Python Code**

```python
from typing import Literal
import workspace.MATH.workflows.template.operator as operator
import workspace.MATH.workflows.round_16.prompt as prompt_custom
from scripts.async_llm import create_llm_instance
import weave

DatasetType = Literal["HumanEval", "MBPP", "GSM8K", "MATH", "HotpotQA", "DROP"]

class Workflow:
    def __init__(
        self,
        name: str,
        llm_config,
        dataset: DatasetType,
    ) -> None:
        self.name = name
        self.dataset = dataset
        self.llm = create_llm_instance(llm_config)
        self.custom1 = operator.Custom(self.llm)
        self.custom2 = operator.Custom(self.llm)
        self.custom3 = operator.Custom(self.llm)
        self.custom4 = operator.Custom(self.llm)
        self.custom5 = operator.Custom(self.llm)
        self.programmer = operator.Programmer(self.llm)
        self.refine = operator.Custom(self.llm)
        self.ensemble = operator.ScEnsemble(self.llm)

    @weave.op()
    async def __call__(self, problem: str):
        """Implementation of the workflow"""
        # Use the first custom operator to solve the problem step by step
        custom_response_1 = await self.custom1(input=problem,
        ↪   instruction=prompt_custom.SIMPLE_SOLVER, role="simple_solver_1")

        # Use the second custom operator to solve the problem step by step
        custom_response_2 = await self.custom2(input=problem,
        ↪   instruction=prompt_custom.SIMPLE_SOLVER, role="simple_solver_2")

        # Use the third custom operator to provide an alternative approach to the
        ↪   problem
        custom_response_3 = await self.custom3(input=problem,
        ↪   instruction=prompt_custom.ALTERNATIVE_SOLVER, role="alternative_solver")

        # Use the fourth custom operator to provide a detailed solution outline
        custom_response_4 = await self.custom4(input=problem,
        ↪   instruction=prompt_custom.DETAILED_SOLUTION_OUTLINE,
        ↪   role="detailed_solution_outline")

        # Use the fifth custom operator to provide a comprehensive solution
        custom_response_5 = await self.custom5(input=problem,
        ↪   instruction=prompt_custom.COMPREHENSIVE_SOLUTION,
        ↪   role="comprehensive_solution")

        # Use the programmer operator to analyze the problem and provide a detailed
        ↪   solution
        programmer_response = await self.programmer(problem=problem, analysis="Solve
        ↪   the math problem step by step")

        # Combine all responses into a list for ensemble processing
        solutions = [
            custom_response_1['response'],
            custom_response_2['response'],
            custom_response_3['response'],
            custom_response_4['response'],
            custom_response_5['response'],
            programmer_response['output']
        ]

        # Use the ensemble operator to select the best solution
```

```
ensemble_response = await self.ensemble(solutions=solutions, problem=problem)

# Use the refine operator to ensure clarity and correctness of the output
refined_response = await self.refine(input=ensemble_response['response'],
↪  instruction=prompt_custom.REFINE_SOLUTION, role="refine_solution")

# Return the final output and cost
return refined_response['response'], self.llm.get_usage_summary()["total_cost"]
```

**Corresponding Prompt**

```
SIMPLE_SOLVER = """Please solve the given mathematical problem step by step. Follow
↪  these guidelines:

1. State the problem clearly.
2. Outline the approach and any relevant formulas or concepts.
3. Provide detailed calculations, using LaTeX notation for mathematical expressions.
4. Explain each step of your reasoning.
5. Present the final answer enclosed in \\boxed{} LaTeX notation.
6. Ensure all mathematical notation is in LaTeX format.
"""

ALTERNATIVE_SOLVER = """Please provide an alternative approach to solving the given
↪  mathematical problem. Follow these guidelines:

1. Clearly restate the problem.
2. Identify any different methods or perspectives that could be applied.
3. Provide calculations and reasoning, using LaTeX notation for mathematical
↪  expressions.
4. Ensure clarity and correctness in your explanation.
5. Present the final answer enclosed in \\boxed{} LaTeX notation.
"""

DETAILED_SOLUTION_OUTLINE = """Please provide a detailed outline for solving the given
↪  mathematical problem. Follow these guidelines:

1. Clearly restate the problem.
2. Identify key concepts and theorems relevant to the problem.
3. Outline the steps needed to solve the problem, including any necessary calculations.
4. Ensure that the outline is structured logically and is easy to follow.
5. Use LaTeX notation for any mathematical expressions.
"""

COMPREHENSIVE_SOLUTION = """Please provide a comprehensive solution to the given
↪  mathematical problem. Follow these guidelines:

1. Clearly restate the problem.
2. Explain the mathematical concepts and theorems involved.
3. Provide a detailed, logical progression of steps leading to the solution.
4. Show all calculations using LaTeX notation for mathematical expressions.
5. Present the final answer clearly marked and enclosed in \\boxed{} LaTeX notation.
"""

REFINE_SOLUTION = """Given the mathematical problem and the solutions generated, please
↪  refine the output to ensure clarity and correctness. Follow these guidelines:

1. Review the solutions provided.
2. Ensure all calculations are accurate and clearly presented.
3. Summarize the findings and present the final answer in a clear format.
4. Use LaTeX notation for any mathematical expressions.
5. Ensure the final answer is enclosed in \\boxed{} LaTeX notation.
"""
```

## B.3 HUMANEVAL

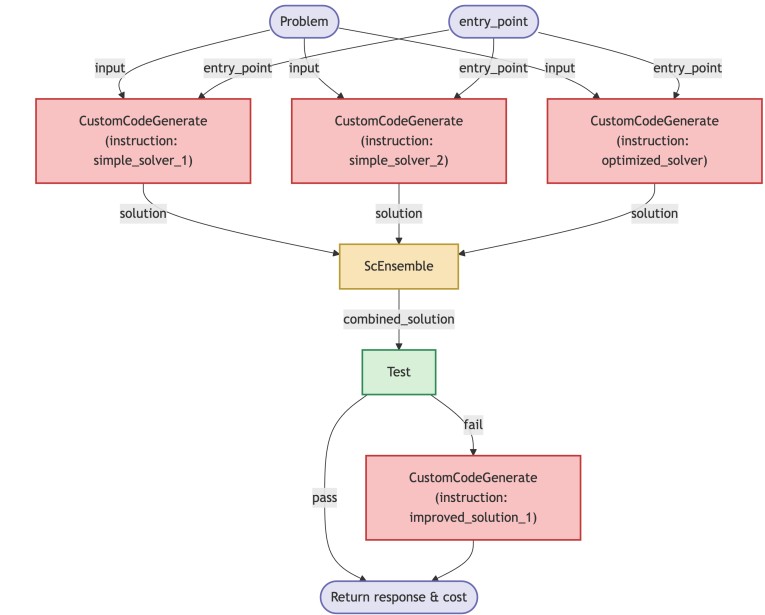

Figure 7: Mermaid diagram for HumanEval.

**Mermaid Code**

```
flowchart TD
    %% Nodes
    PROBLEM([Problem])
    ENTRY_POINT([entry_point])
    C1["CustomCodeGenerate
(instruction: simple_solver_1)"]
    C2["CustomCodeGenerate
(instruction: simple_solver_2)"]
    C3["CustomCodeGenerate
(instruction: optimized_solver)"]
    C4["CustomCodeGenerate
(instruction: improved_solution_1)"]
    ENSEMBLE["ScEnsemble
"]
    T["Test
"]
    RETURN([Return response & cost])

    %% Styles
    classDef CustomOp fill:#d0e1f9,stroke:#4378a2,stroke-width:2px;
    classDef CustomCodeGenerateOp fill:#f9c2c2,stroke:#c23737,stroke-width:2px;
    classDef ScEnsembleOp fill:#f9e4b7,stroke:#b99b37,stroke-width:2px;
    classDef TestOp fill:#d8f0d8,stroke:#2e8b57,stroke-width:2px;
    classDef Interface fill:#e2e2f2,stroke:#6a6ab2,stroke-width:2px;

    %% Assign classes
    class PROBLEM Interface
    class ENTRY_POINT Interface
    class C1 CustomCodeGenerateOp
    class C2 CustomCodeGenerateOp
    class C3 CustomCodeGenerateOp
    class C4 CustomCodeGenerateOp
    class ENSEMBLE ScEnsembleOp
    class T TestOp
    class RETURN Interface

    %% Flow (arrows show data relationships)
    PROBLEM --> |input|C1
    PROBLEM --> |input|C2
    PROBLEM --> |input|C3
    ENTRY_POINT --> |entry_point|C1
    ENTRY_POINT --> |entry_point|C2
    ENTRY_POINT --> |entry_point|C3
    C1 --> |solution|ENSEMBLE
    C2 --> |solution|ENSEMBLE
    C3 --> |solution|ENSEMBLE
    ENSEMBLE --> |combined_solution|T
```

```
      T --> |fail|C4
      T --> |pass|RETURN
      C4 --> RETURN
```

**Python Code**

```python
from typing import Literal
import workspace.HumanEval.workflows.template.operator as operator
import workspace.HumanEval.workflows.round_5.prompt as prompt_custom
from scripts.async_llm import create_llm_instance
import weave

DatasetType = Literal["HumanEval", "MBPP", "GSM8K", "MATH", "HotpotQA", "DROP"]

class Workflow:
    def __init__(
        self,
        name: str,
        llm_config,
        dataset: DatasetType,
    ) -> None:
        self.name = name
        self.dataset = dataset
        self.llm = create_llm_instance(llm_config)
        self.custom = operator.Custom(self.llm)
        self.custom_code_generate = operator.CustomCodeGenerate(self.llm)
        self.sc_ensemble = operator.ScEnsemble(self.llm)
        self.test = operator.Test(self.llm)

    async def __call__(self, problem: str, entry_point: str):
        """
        Implementation of the workflow
        Custom operator to generate multiple solutions and select the best one.
        """
        # Generate initial solutions using custom code generation
        solution_response_1 = await self.custom_code_generate(problem=problem,
        ↪  entry_point=entry_point, instruction=prompt_custom.CODE_GENERATE_PROMPT)
        solution_1 = solution_response_1['response']

        solution_response_2 = await self.custom_code_generate(problem=problem,
        ↪  entry_point=entry_point, instruction=prompt_custom.CODE_GENERATE_PROMPT)
        solution_2 = solution_response_2['response']

        # Generate an optimized solution
        optimized_solution_response = await self.custom_code_generate(problem=problem,
        ↪  entry_point=entry_point,
        ↪  instruction=prompt_custom.OPTIMIZED_CODE_GENERATE_PROMPT)
        optimized_solution = optimized_solution_response['response']

        # Combine solutions using ensemble method
        ensemble_response = await self.sc_ensemble(solutions=[solution_1, solution_2,
        ↪  optimized_solution], problem=problem)
        combined_solution = ensemble_response['response']

        # Test the combined solution
        test_result = await self.test(problem=problem, solution=combined_solution,
        ↪  entry_point=entry_point)

        # If the solution fails the test, improve the code
        if not test_result['result']:
            improved_solution_response = await self.custom(input=problem,
            ↪  instruction=prompt_custom.IMPROVE_CODE_PROMPT)
            improved_solution = improved_solution_response['response']
            # Test the improved solution
            test_result = await self.test(problem=problem, solution=improved_solution,
            ↪  entry_point=entry_point)
            combined_solution = improved_solution if test_result['result'] else
            ↪  combined_solution  # Use improved solution if it passes tests

        return combined_solution, self.llm.get_usage_summary()["total_cost"]
```

**Corresponding Prompt**

```
IMPROVE_CODE_PROMPT = """
The previous solution failed some test cases in the HumanEval benchmark. Please conduct
↪  a thorough analysis of the problem statement, identifying all edge cases and
↪  potential pitfalls. Then, provide an improved solution that not only fixes the
↪  issues but also optimizes performance and adheres to industry-standard coding
↪  practices. Ensure your revised code includes clear, concise comments that explain
↪  your logic and design choices, and that it robustly handles all specified
↪  requirements.
"""

CODE_GENERATE_PROMPT = """
Generate a Python function to solve the given problem. Ensure the function name matches
↪  the one specified in the problem. Include necessary imports. Use clear variable
↪  names and add comments for clarity.

Problem:
{problem}

Function signature:
{entry_point}

Generate the complete function below:
"""

OPTIMIZED_CODE_GENERATE_PROMPT = """
Based on previous attempts and their outcomes, generate an optimized Python function to
↪  solve the given problem. Focus on improving efficiency, readability, and
↪  robustness. Ensure the function name matches the one specified in the problem,
↪  include necessary imports, and use clear variable names with comments for clarity.

Problem:
{problem}

Function signature:
{entry_point}

Generate the complete function below:
"""

ENSEMBLE_PROMPT = """
You have multiple solutions generated for the problem. Your task is to analyze these
↪  solutions and select the one that is most likely to be correct based on their
↪  similarities and performance. Ensure that the selected solution is robust and
↪  handles all edge cases effectively.
"""
```

### B.4 MBPP

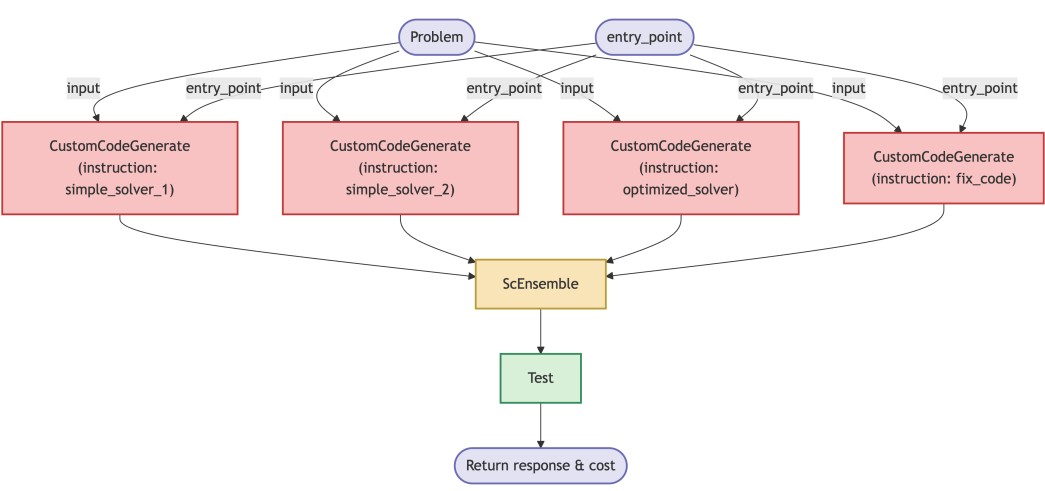

Figure 8: Mermaid diagram for MBPP.

**Mermaid Code**

```
flowchart TD
%% Nodes
PROBLEM([Problem])
ENTRY_POINT([entry_point])
C1["CustomCodeGenerate
(instruction: simple_solver_1)"]
C2["CustomCodeGenerate
(instruction: simple_solver_2)"]
C3["CustomCodeGenerate
(instruction: optimized_solver)"]
C4["CustomCodeGenerate
(instruction: fix_code)"]
ENSEMBLE["ScEnsemble
"]
T["Test
"]
RETURN([Return response & cost])

%% Styles
classDef CustomOp fill:#d0e1f9,stroke:#4378a2,stroke-width:2px;
classDef CustomCodeGenerateOp fill:#f9c2c2,stroke:#c23737,stroke-width:2px;
classDef ScEnsembleOp fill:#f9e4b7,stroke:#b99b37,stroke-width:2px;
classDef DecisionOp fill:#ffffff,stroke:#444444,stroke-width:1px,stroke-dasharray:2 2;
classDef TestOp fill:#d8f0d8,stroke:#2e8b57,stroke-width:2px;
classDef Interface fill:#e2e2f2,stroke:#6a6ab2,stroke-width:2px;

%% Assign classes
class PROBLEM Interface
class ENTRY_POINT Interface
class C1 CustomCodeGenerateOp
class C2 CustomCodeGenerateOp
class C3 CustomCodeGenerateOp
class C4 CustomCodeGenerateOp
class ENSEMBLE ScEnsembleOp
class T TestOp
class RETURN Interface

%% Flow (arrows show data relationships)
PROBLEM --> |input|C1
PROBLEM --> |input|C2
PROBLEM --> |input|C3
PROBLEM --> |input|C4
ENTRY_POINT --> |entry_point|C1
ENTRY_POINT --> |entry_point|C2
ENTRY_POINT --> |entry_point|C3
ENTRY_POINT --> |entry_point|C4
C1 --> ENSEMBLE
```

```
C2 --> ENSEMBLE
C3 --> ENSEMBLE
C4 --> ENSEMBLE
ENSEMBLE --> T
T --> RETURN
```

**Python Code**

```python
from typing import Literal
import workspace.MBPP.workflows.template.operator as operator
import workspace.MBPP.workflows.round_8.prompt as prompt_custom
from scripts.async_llm import create_llm_instance
import weave

DatasetType = Literal["HumanEval", "MBPP", "GSM8K", "MATH", "HotpotQA", "DROP"]

class Workflow:
    def __init__(
        self,
        name: str,
        llm_config,
        dataset: DatasetType,
    ) -> None:
        self.name = name
        self.dataset = dataset
        self.llm = create_llm_instance(llm_config)
        self.custom = operator.Custom(self.llm)
        self.custom_code_generate = operator.CustomCodeGenerate(self.llm)
        self.sc_ensemble = operator.ScEnsemble(self.llm)
        self.test = operator.Test(self.llm)

    @weave.op()
    async def __call__(self, problem: str, entry_point: str):
        """
        Implementation of the workflow
        Custom operator to generate anything you want.
        But when you want to get standard code, you should use custom_code_generate
        ↪  operator.
        """
        # Generate two solutions using different instructions
        solution_1 = await self.custom_code_generate(problem=problem,
        ↪  entry_point=entry_point, instruction=prompt_custom.CODE_GENERATE_PROMPT)
        solution_2 = await self.custom_code_generate(problem=problem,
        ↪  entry_point=entry_point, instruction=prompt_custom.CODE_GENERATE_PROMPT)
        # Generate an optimized solution
        optimized_solution = await self.custom_code_generate(problem=problem,
        ↪  entry_point=entry_point, instruction=prompt_custom.OPTIMIZED_CODE_PROMPT)
        # Attempt to fix the code if necessary
        fix_attempt = await self.custom_code_generate(problem=problem,
        ↪  entry_point=entry_point, instruction=prompt_custom.FIX_CODE_PROMPT)

        # Use ensemble to select the best solution
        ensemble_response = await self.sc_ensemble(solutions=[solution_1['response'],
        ↪  solution_2['response'], optimized_solution['response'],
        ↪  fix_attempt['response']], problem=problem)

        # Test the selected solution
        test_response = await self.test(problem=problem,
        ↪  solution=ensemble_response['response'], entry_point=entry_point)

        # Return the final response and cost
        return test_response['solution'], self.llm.get_usage_summary()["total_cost"]
```

**Corresponding Prompt**

```
CODE_GENERATE_PROMPT = """
Generate a Python function to solve the given problem. Ensure the function name matches
↪  the one specified in the problem. Include necessary imports. Use clear variable
↪  names and add comments for clarity.

Problem:
{problem}

Function signature:
```

```
{entry_point}

Generate the complete function below:
"""

OPTIMIZED_CODE_PROMPT = """
Based on previous attempts, generate an optimized Python function to solve the given
↪  problem. Ensure the function name matches the one specified in the problem. Include
↪  necessary imports, and focus on improving performance and readability. Use clear
↪  variable names and add comments for clarity.

Problem:
{problem}

Function signature:
{entry_point}

Generate the complete function below:
"""

FIX_CODE_PROMPT = """
The provided solution failed to pass the tests. Please analyze the error and fix the
↪  code. Ensure the function name and signature remain unchanged. If necessary, add or
↪  modify imports, correct logical errors, and improve the implementation.

Problem:
{input}

Provide the corrected function below:
"""
```

## C  COMMON TYPES OF PYTHON SCRIPT FAILURES

In this section we will list the type of failures that python script often arise, and with some examples to demonstrate that.

1. Python's flexibility often misleads the LLM into producing superficially reasonable but unreliable control logic
    (a) Unreliable if conditions, the LLM's decision on whether a condition should trigger is often incorrect, since you can guarantee the behavior of LLM's output.
    (b) Using a single prompt to drive a for loop over many iterations offers essentially no benefit when the temperature is set to 0.
2. Incorrect instance initialization
    (a) Parameters may be incorrect or missing.
3. Importing or referencing nonexistent modules/repositories

Here are some detailed Python script examples; we truncate irrelevant code segments and highlight violation points using comments placed before the corresponding snippets.

### C.1  UNRELIABLE CONTROL LOGIC FROM PYTHON'S FLEXIBILITY

**Unreliable if conditions**

```
...
# This `if` condition depends on the LLM's output, but the prompt does not specify the
↪  expected output format.
for solution in solutions:
    verify = await self.programmer(
        problem=problem,
        analysis=f"Verify if this solution is mathematically correct: {solution}"
    )
    if verify['output'].lower().startswith('correct'):
        verified_solutions.append(solution)
...
```

**Meaningless for loop**

```
...
# Since all iterations use the same prompt and the default temperature is 0 (as in
↪  AFLOW), this for-loop is meaningless and only wastes tokens.
solutions = []
for _ in range(5):
    solution = await self.custom(input=problem,
    ↪  instruction=prompt_custom.GENERATE_SOLUTION_PROMPT)
    solutions.append(solution['response'])
...
```

## C.2  INCORRECT INSTANCE INITIALIZATION

**Missing paramters**

```
...
# When initializing different nodes like programmer or sc_ensemble, they all require
↪  the `self.llm` parameter, but it gets ignored.
self.name = name
self.dataset = dataset
self.llm = create_llm_instance(llm_config)
self.custom = operator.Custom(self.llm)
self.programmer = operator.Programmer()
self.sc_ensemble = operator.ScEnsemble()
...
```

**Passing incorrect parameters**

```
...
# When initializing the LLM, incorrect instructions may be passed, since these
↪  instructions are provided only at inference time.
self.name = name
self.dataset = dataset
self.llm = create_llm_instance(llm_config)
self.custom1 = operator.Custom(self.llm, instruction=prompt_custom.SOLVE_PROMPT) #
↪  shouldn't pass instruction
self.custom2 = operator.Custom(self.llm, instruction=prompt_custom.VERIFY_PROMPT)
self.sc_ensemble = operator.ScEnsemble(self.llm)
...
```

## C.3  IMPORTING OR REFERENCING NONEXISTENT MODULES/REPOSITORIES

**Importing unsupported packages**

```
...
# When generating the script, the LLM may import incorrect or disallowed packages.
from typing import Literal
import pandas as pd # this repo is not allowed to import
import workspace.MATH.workflows.template.operator as operator
import workspace.MATH.workflows.round_16.prompt as prompt_custom
from scripts.async_llm import create_llm_instance
...
```

# D  THE ERRORS IN MERMAID CODE AND THE FREQUENCY

In this section, we summarize the retry error reasons observed during the four benchamrk and earch benchmark will evolve 20-round. We detect five types of errors, as listed in A.2. Their frequencies are shown in the following table.

Table 5: Error Frequency

| Error Type | W1 | W2 | W3 | W4 | W5 |
|---|---|---|---|---|---|
| **Violation counts over 20×4 rounds** | 0 | 4 | 0 | 2 | 3 |

Here are some examples of incorrectly generated Mermaid workflows. We have rendered the workflows as images for clearer illustration.

## D.1  GENERATE A NODE THAT CANNOT CONNECT PROPERLY TO THE INTERFACE

This will be detected by W2 (No isolated nodes are allowed).

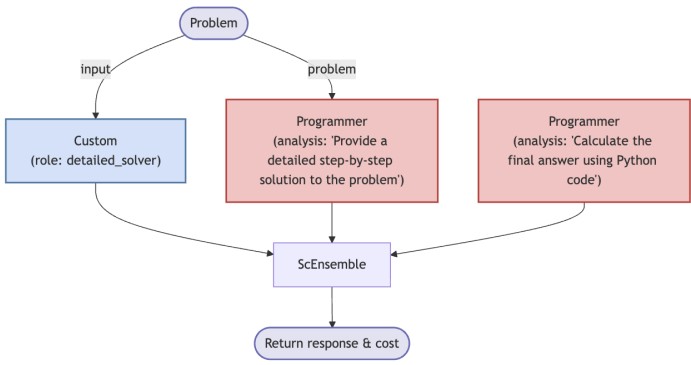

Figure 9: The right node isn't connected to the Problem Interface node.

## D.2  CREATE NEW TYPE OF NODE

This will be detected by W4 (Do not create new types).

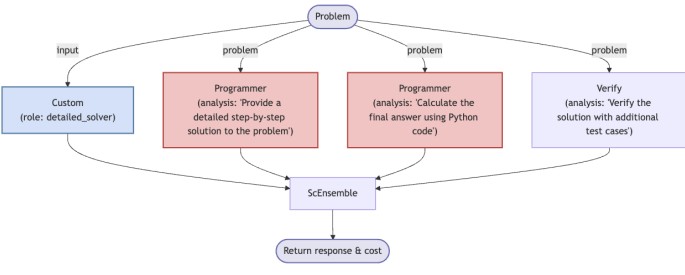

Figure 10: The right node creates a new type called Verify, which is not allowed.

## D.3  ONLY ONE NODE TO INPUT TO ENSEMBLE NODE

This will be detected by W5 (Ensemble nodes must have at least two inputs).

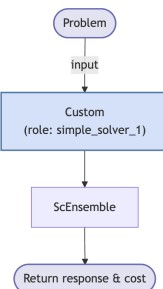

Figure 11: The ensemble node only takes one input, which is not a reasonable evolutionary process.

# E    FUTURE WORK

Using Mermaid can lead to better and more stable update steps, but in its current form MermaidFlow lacks certain representational capabilities, such as expressing if-conditions or for-loops. We can address this by adding more comprehensive node types to support such constructs, or by fine-tuning a model specifically trained to generate valid Mermaid scripts. With a Mermaid checker or compiler, we can more easily build training datasets or use them as sources of reward signals during post-training.

Another necessary step is to create a rule-based Mermaid-to-Python converter. In the current version, the Mermaid-to-Python translation relies on an LLM, which can lead to issues similar to those seen in direct Python script generation. Because Mermaid has a simple syntax, it should be straightforward to perform reliable transformations into valid Python LangGraph scripts. This step will significantly improve the robustness of MermaidFlow.

Another interesting direction worth exploring is extending the current task setting into a retrieval-based paradigm. Specifically, after generating a Mermaid graph, it can be stored directly in its graph form; when encountering a new task with a structurally similar requirement, the system can retrieve relevant workflows as references. Such a mechanism would enable efficient storage and reuse of a collection of verified workflows, allowing them to be retrieved when needed and thereby providing additional performance gains.

# F    LLMS USAGE

In the preparation of this paper, LLMs are utilized solely for drafting and proofreading purposes. All content produced by LLMs is thoroughly reviewed and edited by the authors. The authors take full responsibility for the final content of the paper.

