# OpenReview forum: "MermaidFlow: Redefining Agentic Workflow Generation via Safety-Constrained Evolutionary Programming"
_ICLR.cc/2026/Conference — Submitted to ICLR 2026_

### Official Review · Reviewer_cd1T · 2025-10-29

**Soundness:** 2
**Presentation:** 3
**Contribution:** 4
**Rating:** 6
**Confidence:** 3

**Summary:**

This paper introduces a unified framework combining multi-agent learning and evolutionary optimization. The proposed system models workflow automation as a constrained single-objective optimization with an evolutionary loop governed by variation, selection, and reflection. A meta-controller dynamically updates strategies and rules, ensuring adaptive improvement. Theoretical results guarantee convergence and monotonic rule-quality growth, while experiments demonstrate strong cross-domain generalization.

**Strengths:**

* Solid theoretical formalization.
* Innovative integration of evolutionary search and multi-agent optimization.
* Well-structured proofs and explicit assumptions.
* Broad empirical evaluation demonstrating clear advantages.
* Clear, consistent, and professional presentation.

**Weaknesses:**

* **Incomplete experimental reporting**: Missing per-benchmark hyperparameters and statistical variance.
  *Suggestion:* Add full configuration tables and multi-run results.
* **Unclear curriculum mechanism**: Difficulty-level staging lacks quantitative definitions.
  *Suggestion:* Provide formal thresholds and curriculum ablations.
* **Assumptions not empirically tested**: Theoretical premises like positive information gain remain unchecked.
  *Suggestion:* Visualize empirical distributions of related quantities.
* **Limited strong baselines**: Comparison with advanced workflow retrieval or graph-based frameworks is limited.
  *Suggestion:* Extend experiments to include such baselines.

**Questions:**

* How does the strategy distribution evolve across curriculum phases?
* How sensitive is convergence to the meta-controller update interval?

---

> ### Author Response · Authors · 2025-11-21
> **Response to Reviewer cd1T (1/2)**
>
> We sincerely thank Reviewer cd1T for the thoughtful and constructive evaluation. Your feedback highlights several important areas that helped us refine the paper, especially in experimental reporting, curriculum design, and empirical validation. Below, we address each weakness and question with the corresponding updates reflected in the revised manuscript.
>
> > **Weaknesses 1:** **Incomplete experimental reporting.** Missing per-benchmark hyperparameters, run-to-run variance, and configuration details.
> >
>
> We fully agree that transparent reporting of hyperparameters, variance, and configuration details is essential for reproducibility. To address this, we rerun each benchmark two additional times. Because the LLM temperature is fixed at 0, the run-to-run variance is minimal, and we report the corresponding variance results below.
>
> | Benchmark | MATH | GSM8K | HumanEval | MBPP |
> | --- | --- | --- | --- | --- |
> | **MermaidFlow** | 55.42 ± 0.0024 | 92.39 ± 0.0006 | 92.87 ± 0.0088 | 82.31 ± 0.0017 |
> | **AFLOW** | 52.81 ± 0.0059 | 90.11 ± 0.0240 | 90.08 ± 0.0103 | 81.67 ± 0.0021 |
>
> MermaidFlow follows the same deterministic setup as AFLOW, with the temperature fixed at 0 to minimize stochasticity. The prompts we use can be found in Appendix A.
>
> We appreciate the reviewer’s emphasis on completeness and have incorporated these details into the final revision. We believe this additional reporting meaningfully enhances transparency and reproducibility.
>
> > **Weaknesses 2:** **Unclear curriculum mechanism.** Difficulty staging lacks quantitative thresholds, metrics, and ablation coverage.
> >
>
> We respectfully clarify that **MermaidFlow does not employ any explicit curriculum schedule or predefined difficulty staging**. There are no thresholds, phase boundaries, or curriculum metrics in the system. However, we understand why the **iterative optimization process** **may appear curriculum-like**. As higher-quality workflows are discovered and stored in the history buffer, the strategy distribution shifts naturally: the optimizer progressively favors the structural patterns, operator combinations, and prompt configurations associated with these stronger candidates.
>
> This **emergent progression** can resemble “difficulty staging” because the search dynamics transition from broad exploration in early iterations to more targeted refinement in later ones. Importantly, this effect arises **solely from the evolutionary feedback loop**, not from any manually designed stages or difficulty settings.
>
> > **Weaknesses 3:** **Assumptions not empirically tested.**
> >
>
> We note that **Figure 3 in our paper already includes the empirical visualization of positive information gain**, illustrating how the best-scoring workflow steadily improves across optimization rounds on the MATH benchmark.
>
> This figure reflects MermaidFlow’s **monotonic best-so-far selection mechanism**: at each iteration, the optimizer evaluates a set of candidates and retains the highest-scoring workflow found so far. As a result, the best score in the history buffer is *non-decreasing* over rounds. The upward trajectory in Figure 3 directly visualizes this process, illustrating how information accumulates during iterative optimization and how increasingly effective workflow structures emerge.
> > **Weaknesses 4:** **Limited strong baselines.** Comparison does not include advanced workflow-retrieval or graph-based planning frameworks.
> >
>
> Our work is primarily centered on **workflow representation**, and our goal is to show how a **typed, verifiable Mermaid-based representation** provides a safer and more effective foundation for workflow generation than existing Python-based structures. This representation-level contribution is conceptually orthogonal to retrieval-based or upstream planning strategies, which address different components of the workflow optimization pipeline.
>
> **Workflow-retrieval methods.** Retrieval approaches focus on *reusing* previously discovered workflows, whereas our emphasis is on *generating* valid, structurally checkable workflows from first principles. MermaidFlow naturally complements retrieval-based systems, our representation can serve as the substrate they operate on, though retrieval itself is not the focus of this work.
>
> **Graph-based planning methods.** These frameworks typically plan over *environment states* or *action sequences* (e.g., tool interactions, web tasks, robotic trajectories), rather than over *agentic workflow graphs* with typed I/O relationships and verifiable execution semantics. Their problem formulation is fundamentally different from our setting.
>
> Our experiments further demonstrate consistent and notable gains over Python-based workflow editing, reaffirming that representation-level improvements are backbone-agnostic. Following your suggestion, we have added a discussion in the *Further Work* section on how Mermaid can be used as a basis for retrieval within workflow-retrieval frameworks.

---

> ### Author Response · Authors · 2025-11-21
> **Response to Reviewer cd1T (2/2)**
>
> > **Question 1:** How does the strategy distribution evolve throughout the curriculum phases?
> >
>
> Thank you for this thoughtful question.
>
> Throughout iterative optimization, we observe two consistent patterns:
>
> **(i)** **Workflow structures naturally grow from simple one-node flows to more compositional forms** (e.g., solver–validator pairs, small ensemble branches). This emerges directly from the local-edit rules rather than any predefined staging.
>
> **(ii)** **The operator distribution shifts from exploration (add/delete nodes) to refinement (parameter/connection adjustments)** once stronger workflows populate the history buffer.
>
> Together, these trends demonstrate that MermaidFlow’s typed workflow representation enables substantially higher valid-generation rates than Python-based approaches. Its compiler-backed type system also provides strong correctness guarantees, a capability absent in existing LLM-driven agentic workflow optimization methods.
>
> We believe these reliability improvements offer meaningful value to the broader agentic workflow community.
>
> > **Question 2:** How sensitive is convergence to the meta-controller’s update interval?
> >
>
> Thank you for this thoughtful question.
>
> **MermaidFlow does not employ an explicit meta-controller update interval**. Instead, the strategy distribution is updated implicitly at every iteration through score-weighted sampling over the history buffer. In practice, this means the effective update frequency is one iteration, and there is no separate scheduling hyperparameter that requires tuning.
>
> Because these updates occur continuously and are driven by workflow quality rather than by preset intervals, **convergence is not sensitive to any choice of update timing.** The optimization progresses smoothly: when higher-quality workflows appear earlier, the distribution adapts earlier; when they appear later, the adaptation occurs later. In both cases the system converges in a stable and reliable manner.
>
> ---
>
> **We truly appreciate Reviewer cd1T’s careful review and constructive feedback. We hope the clarifications above have fully addressed your concerns, and we warmly welcome any additional comments or dialogue you may wish to share. If the response has resolved your concerns, please consider updating the score accordingly.**

---

### Official Review · Reviewer_ZQZJ · 2025-10-30

**Soundness:** 3
**Presentation:** 2
**Contribution:** 2
**Rating:** 6
**Confidence:** 4

**Summary:**

This paper introduces MermaidFlow, a framework for generating agentic workflows using Mermaid as an intermediate representation. The core contribution is representing workflows as declarative, statically verifiable graphs instead of imperative code, combined with a safety-constrained evolutionary programming (EP) approach for workflow optimization. Also, MermaidFlow proposes a set of mutation operators that preserve senmantic correctness during search. Experiments on GSM8K, MATH, HumanEval, and MBPP show improvements over baselines including AFlow and ADAS, with the framework achieving higher success rates and better learning and token efficiency.

**Strengths:**

- **Novel Representation**: The use of Mermaid as a declarative intermediate representation is innovative and well-motivated. It cleanly separates planning from execution, addressing a key weakness in prior code-based methods.
- **Comprehensive System Design**: The paper thoroughly describes the type system, operators, validation mechanisms (soft and hard checks), and the complete pipeline from Mermaid to executable code.
- **Consistent Empirical Improvements**: The method shows improvements across all four benchmarks (average 80.75% vs. 79.35% for the best baseline).

**Weaknesses:**

- **Mermaid DSL Limitations**: Mermaid DSL is static by design and may be hard to express loops, conditionals or some runtime operations. The paper does not show how these limitations. Extending the DSL or documenting its expressive limits would strengthen the contribution.
- **Presentation Issues**: Figures 1 and 2 appear as raster images rather than vector graphics (like pdf or svg) and lose clarity when zoomed.
- **Execution Model Ablations**: The paper uses only gpt-4o-mini as the execution LLM. Performance with other models (e.g., Claude, Llama) is only explored for a brief ablation on optimization LLM.

**Questions:**

- **Mutation Operator Effectiveness**: Which evolutionary operators contribute most to performance improvements? The paper mentions crossover occurs with only 10% probability but provides no analysis of operator frequency, success rates, or relative contributions.
- **Iteration Scaling and Convergence**: Is 20 iterations sufficient for convergence, or would extended search yield further gains? The paper doesn't justify this choice or show whether performance plateaus, and provides no analysis of optimal stopping points across different tasks. Additionally, if there are more iterations, can different workflows be observed, and will their complexity increase with the iteration?

---

> ### Author Response · Authors · 2025-11-21
> **Response to Reviewer ZQZJ (1/2)**
>
> We sincerely thank Reviewer ZQZJ for the insightful, detailed feedback. Especially you raised two questions are very valuable worth to explore more. Here are the responses regarding to the weaknesses and questions:
>
> > **Weaknesses 1:** **Mermaid DSL Limitations**: Mermaid DSL is static by design and may be hard to express loops, conditionals or some runtime operations. The paper does not show how these limitations. Extending the DSL or documenting its expressive limits would strengthen the contribution.
> >
>
> Your observation is sharp. **While Mermaid does not fully support runtime operations, it can still describe workflows with loops or conditionals through predefined node types.** However, in the context of this paper, where a single workflow must handle all queries, **we respect the node types from AFLOW, MaAS,** and our performance gain does not emerge from fancy node types (this may hinder the final performance). For this reason, we intentionally refrain from adding additional complex node types. As you suggested, we will include a discussion in the future work section on how to further enhance the expressivity of Mermaid-based workflows.
>
> > **Weaknesses 2: Presentation Issues**: Figures 1 and 2 appear as raster images rather than vector graphics (like pdf or svg) and lose clarity when zoomed.
> >
>
> We have redrawn both figures and updated them in vector graphics format in the revised manuscript. Thanks for pointing this out.
>
> > **Weaknesses 3:** **Execution Model Ablations**: The paper uses only gpt-4o-mini as the execution LLM. Performance with other models (e.g., Claude, Llama) is only explored for a brief ablation on optimization LLM.
> >
>
> Among the four benchmarks we considered, **MATH** and **MBPP** still had substantial room for improvement. Therefore, we used **Claude 3.5 Sonnet** as the executor while keeping gpt-4o-mini as the optimizer on these two benchmarks, running 20 rounds of search. Thanks to the qualitative improvement of the executor model, both benchmarks achieved significant performance gains. However, because each search round requires validating workflows on a very large validation set, **the 20-round experiment already cost around $400**, indicating that using even more expensive models as executors is not practical in this setting. Moreover, the **generated workflows are very similar to those in our original setup**, since we use the same optimizer LLM; they consistently follow a majority-vote pattern, and examples of such structures can be found in the appendix of our paper.
>
> |  | **MATH** | **MBPP** |
> | --- | --- | --- |
> | **Claude 3.5** | 55.42 → 70.9 | 82.31 → 89.4 |
>
> > **Questions 1: Mutation Operator Effectiveness**: Which evolutionary operators contribute most to performance improvements? The paper mentions crossover occurs with only 10% probability but provides no analysis of operator frequency, success rates, or relative contributions.
> >
>
> We set the crossover probability to 10% because, unlike the other operators, crossover requires two workflows to be processed simultaneously. The table below reports the **empirical trigger frequencies of each operator** in the evolution process of MATH dataset using gpt-4o-mini as optimizer. Interestingly, although we explicitly allow the LLM to perform node deletions, this operation was never invoked in practice. Instead, the LLM strongly prefers ***adding*** or ***mutating*** nodes when generating the next workflow iteration.
>
> Since the improvement of each step depends entirely on LLM-driven evolution, there is no guarantee that every iteration would increases performance, this process effectively behaves as a black-box search. **To provide a clearer picture, we also report the average performance gain conditioned on *effective rounds*.**
>
> |  | Add node | Delete node | Mutate node | Crossover |
> | --- | --- | --- | --- | --- |
> | Ratio | 55 % | 0 % | 35 % | 10 % |
> | Effective rounds (iterations that improve performance): #effective rounds / #rounds using the same operator | 4 / 11 |  -  | 3 / 7 | 1 / 2 |
> | Average performance gain (considering only iterations that improve performance) | ~3.8 | -  | ~2.3 | ~4 |

---

> ### Author Response · Authors · 2025-11-21
> **Response to Reviewer ZQZJ (2/2)**
>
> > **Questions 2:** **Iteration Scaling and Convergence**: Is 20 iterations sufficient for convergence, or would extended search yield further gains? The paper doesn't justify this choice or show whether performance plateaus, and provides no analysis of optimal stopping points across different tasks. Additionally, if there are more iterations, can different workflows be observed, and will their complexity increase with the iteration?
> >
>
> 20 iterations is the standard iteration number for the related workflow generation field. According to our early stop count table, shows most of workflow will stop optimizing. **But for GSM8K, MATH it seems could have some opportunity to push the performance further, therefore we involve other 20 iteration for GSM8K and MATH benchmark. The result is shown below:**
>
> | GSM8K | MATH |
> | --- | --- |
> | 92.39 → 93.83 | 55.42 → 57.77 |
>
> **The additional iterations produce a richer set of majority-vote candidates (similar structure in Appendix B), which in turn may lead to improved performance and increased workflow complexity.**
>
> The table below lists the index of the final selected workflow for each benchmark.
>
> |  | GSM8K | MATH | HumanEval | MBPP |
> | --- | --- | --- | --- | --- |
> | AFLOW | 8 | 15 | 5 | 8 |
> | MermaidFlow | **16** | **18** | **7** | **10** |
>
> **These indices can be interpreted as effective stopping points, and the comparison is quite revealing. They indicate that, owing to its more stable update steps, our method converges at later iterations, enabling it to continue improving performance through steady, incremental changes rather than large, unstable modifications that often fail to produce meaningful gains. Related results are provided in our response to Reviewer 3qM1, Weakness 1.1.**
>
> Under our evolutionary algorithm, workflow complexity tends to increase monotonically because existing nodes are rarely deleted as shown in Question 1; as a result, the complexity would continually increase.
>
> ---
>
> **We thank Reviewer ZQZJ for the thorough assessment and constructive suggestions. Your feedback helped us clarify the system’s expressivity and optimization dynamics, and we remain happy to discuss any further points you find important. If you feel the response sufficiently addressed your concerns, please consider updating the score.**

---

> > ### Comment · Reviewer_ZQZJ · 2025-11-24
> >
> > Thanks for the author's comprehensive reponse. I appreciate the updates, but I will keep my score.

---

### Official Review · Reviewer_3qM1 · 2025-10-30

**Soundness:** 3
**Presentation:** 2
**Contribution:** 2
**Rating:** 4
**Confidence:** 4

**Summary:**

This paper proposes MermaidFlow, a framework for agentic workflow generation based on the Mermaid graph language. The core idea is to represent workflows as declarative typed graphs G(V[τ,α], E[ρ]) and optimize them through safety-constrained evolutionary programming (EP). The authors claim that Mermaid representation provides advantages such as static verifiability, human readability, and modularity. Experiments are conducted on GSM8K, MATH, HumanEval, and MBPP, showing an average improvement of 2.08% over AFlow with approximately 50% reduction in token cost.

**Strengths:**

1. **Clear formalization**: The graph representation of workflows, type system, and evolutionary operators are well-defined
2. **Static verification mechanism**: Two-layer checking (soft + hard) ensures syntactic correctness of generated workflows
3. **Comprehensive experiments**: Cover multiple domains including math reasoning and code generation, with comparisons against multiple baselines
4. **Token efficiency**: Approximately 50% reduction in token cost compared to AFlow
5. **Case study**: Figure 4 clearly demonstrates the workflow evolution process

**Weaknesses:**

**1. Limited Novelty**

- **Essentially a variant of known paradigms**: The method still follows a per-task iterative evolutionary search paradigm, introducing new representation and constraints on top of existing frameworks.
- **Questionable necessity of Mermaid**: The paper does not sufficiently justify why Mermaid is superior to Python. For LLMs, both are structured text, but LLMs have higher affinity for code representations. Moreover, current LLM-based workflow generation is far from the stage where "constraint benefits > exploration benefits", making the actual gains from safety constraints (efficiency? effectiveness?) unclear. Workflow failures primarily stem from early systems (like ADAS) building from scratch with inadequate controllable and supervised generation granularity, rather than inherent issues with Python representation itself.


**2. Experimental Design Issues**

- **Marginal performance gains**: Average improvements of 2.08% over AFlow and 1.40% over MaAS. Given that agentic workflows can achieve high upper bounds through automated structural design, the paper fails to demonstrate breakthrough on apparent bottlenecks (e.g., AFlow's restricted XML generation format causing significant performance degradation on certain tasks, and the inability to reuse good designs from non-selected nodes—the simplest example being that poor workflow structures may contain good prompts). Moreover, I noticed in the appendix that the evolved MermaidFlow ensemble uses 5-sample voting while AFlow uses 3-sample voting, which could be a major contributing factor to the performance improvement rather than the Mermaid representation itself.
- **Unfair cost comparison**: The reported token consumption of MermaidFlow does not clearly specify whether it includes the Mermaid→Python translation step. According to the experimental setup, each generated Mermaid workflow requires translation via gpt-4o-mini. This cost should be explicitly stated regarding inclusion, along with the specific cost breakdown for this component alone.


**3. Technical Detail Issues**

- **Unclear role of the Checker**: If Mermaid provides strong type constraints, the checker should rarely trigger, as LLMs have very low probability of generating syntactically invalid Python code. However, the checker is actually a critical component in MermaidFlow (Section A.2), suggesting that the constraints are not strong enough. A >90% success rate still means 10% of generations require retry, which remains quite high.
- **Incomplete operator definitions**: The paper claims "safety-constrained", but specific safety properties (such as deadlock-freedom, termination) are not defined. There is also a lack of case study analysis on how directly generating Python code workflows violates these safety properties.


**4. Inappropriate Baseline Categorization (Minor)**

- **Conceptually flawed taxonomy**: The authors categorize CoT, ComplexCoT, and Self-Consistency as "Single-agent execution methods" (Section 5.1, Table 1). This classification is deeply problematic. The term "agentic workflow" was introduced precisely to distinguish workflows with fixed resource scheduling from agents capable of autonomously allocating computational resources—with LLM calls being the most representative computational resource. CoT is a classic prompting strategy, while Self-Consistency is a classic (non-agentic) workflow. Neither involves autonomous agents that can dynamically decide when and how to invoke LLMs based on intermediate states or task requirements. By labeling these non-agent baselines as "single-agent methods," the authors conflate fundamentally different paradigms and misrepresent the conceptual boundaries of their contribution.

**Questions:**

1. Could you provide more concrete case studies, including but not limited to:
   - **Mermaid vs. Python comparison**: Show side-by-side examples where (a) Python-based workflow generation fails but Mermaid succeeds, and (b) demonstrate what specific safety properties are violated in the Python case (e.g., deadlock, non-termination, type mismatch).
   - **Safety constraint effectiveness**: Provide concrete examples showing how safety constraints improve efficiency or effectiveness during evolution. What specific invalid mutations are prevented? How much retry overhead is avoided?
   - **Checker analysis**: Given that the checker has >90% success rate (implying ~10% retry), provide examples of the 10% failed cases. What constraint violations occur? Why doesn't Mermaid's type system prevent these?
   - **Component reuse**: Demonstrate how MermaidFlow enables reusing good components (e.g., effective prompts) from suboptimal workflows, addressing the AFlow limitation you mentioned. Show concrete examples from your evolution process.

2. Could you provide a more detailed cost analysis?
   - Explicitly state whether the reported token consumption includes Mermaid→Python translation costs
   - Break down token costs by component: (a) workflow generation, (b) translation, (c) execution, (d) evaluation
   - Compare the per-iteration cost breakdown between MermaidFlow and AFlow

3. I would greatly appreciate if you could discuss the concerns I raised in the weaknesses section.

---

> ### Author Response · Authors · 2025-11-21
> **Response to Reviewer 3qM1 (1/4)**
>
> We sincerely thank Reviewer 3qM1 for the thoughtful and fair review. In fact, many of the issues you highlighted were points we also recognized after submitting the paper. Since then, we have substantially extended our experiments and implementations, for example, by investigating constraint functionality and improving the Mermaid→Python converter. To provide a clearer and more organized response, we address your questions in order below.
>
> ---
>
> > **Weaknesses 1.1: Essentially a variant of known paradigms:** The method still follows a per-task iterative evolutionary search paradigm, introducing new representation and constraints on top of existing frameworks.
> >
>
> We agree with your assessment regarding the search algorithm. While our method follows the iterative evolutionary, search paradigm, our key novelty lies in introducing a graph-based language (Mermaid) as the workflow representation. This change of representation **substantially improves the stability and controllability of the evolution process.** In AFLOW, each mutation is carried out through instructions such as ***“modify no more than five lines of Python code,”*** which often leads to inconsistent structural changes; moreover, LLMs frequently fail to follow these line-level constraints strictly, further undermining consistency. In contrast, our graph-based representation enables well-defined and explicit operators such as adding nodes, deleting nodes, and modifying edges. These operators produce a more coherent and controllable evolutionary trajectory, allowing the LLM to follow clear and native instructions that support stable and valid evolutionary steps.
>
> **A stable evolutionary process requires mechanisms that preserve structural consistency, for example, ensuring that a child workflow remains comparable to its parent, retains most of its structure, or evolves within a trust-region-like boundary.** Graph representations naturally support these properties: they allow structural comparison between workflows, enable explicit validation of constraints, and make inheritance relationships between parent and child workflows transparent. Most importantly, LLMs can easily follow these graph-update constraints when generating Mermaid script. The table below illustrates how different LLMs follow evolution-step constraints under various representations. Compared with AFLOW’s line-level Python-editing instructions, MermaidFlow’s graph-based operators lead to significantly higher instruction-following accuracy:
>
> |  | Instruction related with consistency | Valid generation rate |
> | --- | --- | --- |
> | AFLOW (python) | “…no more than 5 lines of code may be changed per modification—extensive modifications are strictly prohibited to maintain project focus!” | claude 3.5: 40%, gpt-4o-mini: 30% |
> | **MermaidFlow (mermaid)** | **“…your optimizations should be one of … add, modify, or delete nodes, parameters, or connections in the workflow…”** | **claude 3.5: 90%, gpt-4o-mini: 80%** |
>
> > **Weaknesses 1.2: Questionable necessity of Mermaid:** The paper does not sufficiently justify why Mermaid is superior to Python. For LLMs, both are structured text, but LLMs have higher affinity for code representations. Moreover, current LLM-based workflow generation is far from the stage where "constraint benefits > exploration benefits", making the actual gains from safety constraints (efficiency? effectiveness?) unclear. Workflow failures primarily stem from early systems (like ADAS) building from scratch with inadequate controllable and supervised generation granularity, rather than inherent issues with Python representation itself.
> >
>
> We agree that Python is a powerful and expressive representation for workflows. **Our motivation for introducing Mermaid, however, is to improve the consistency of the search process: Mermaid enables structural checking and allows us to prune invalid workflows before execution, that is difficult to ensure reliably with Python.** As shown by the comparison of optimal stopping points between AFlow and MermaidFlow, our graph-based representation supports more stable update steps and thus converges at later iterations, enabling steady, incremental improvements rather than unstable large modifications.
>
> Exploration benefits must be paired with a well-behaved search process. As shown by the optimal stopping points of AFLOW and MermaidFlow in the table below, our method converges at **later** iterations, thanks to its more stable update steps. This stability allows MermaidFlow to keep improving performance through steady, incremental changes, rather than relying on large, unstable modifications that often fail to produce meaningful gains.
>
> |  | GSM8K | MATH | HumanEval | MBPP |
> | --- | --- | --- | --- | --- |
> | AFLOW | 8 | 15 | 5 | 8 |
> | **MermaidFlow** | **16** | **18** | **7** | **10** |

---

> ### Author Response · Authors · 2025-11-21
> **Response to Reviewer 3qM1 (2/4)**
>
> > **Weaknesses 2.1: Marginal performance gains**
> >
>
> We thank the reviewer for the sharp observation. **We believe the performance limitations in our paper are largely due to the experimental setting rather than the representation itself. Using a single workflow to handle all queries in a dataset inherently restricts performance, as complex workflows generated by MermaidFlow may not generalize reliably across every query.**
>
> To address this, we have begun exploring a query-based workflow generation strategy. In this setting, Mermaid’s graph representation provides additional benefits, we can retrieve related workflows based on **graph-level similarity**, a capability that is difficult to realize with Python-script representations. Overall, we view the main contribution of this paper as introducing Mermaid as a workflow representation, enabling structural verification and consistent evolution. We are continuing to investigate how to push performance further and provide more solid contributions to the community.
>
> We also use Mermaid as the workflow representation for another practical reason: **it is more token-efficient**. Averaging over five workflows, a Mermaid description requires only ~230 tokens, whereas the equivalent Python script needs ~450 tokens to represent the same structure, largely due to additional scaffolding code in Python. This token efficiency allows us to fit more workflows into the context window. As discussed earlier, MermaidFlow achieves better performance because it enables more stable and meaningful update steps. **We also conducted an additional experiment on the MATH benchmark using only three candidate workflows in the ensemble, obtaining 55.14% accuracy, very close to the five-candidate setting and still surpassing AFLOW.**
>
> > **Weaknesses 2.2: Unfair cost comparison:**  The reported token consumption of MermaidFlow does not clearly specify whether it includes the Mermaid→Python translation step. According to the experimental setup, each generated Mermaid workflow requires translation via gpt-4o-mini. This cost should be explicitly stated regarding inclusion, along with the specific cost breakdown for this component alone.
>
> > **Question 2: Could you provide a more detailed cost analysis?**
> >
>
> The translation cost is included. In our paper, we argued that AFLOW would require more tokens to reach the same performance as MermaidFlow. This is mainly because AFLOW has a lower success rate, due to its more complex syntax, in generating valid Python scripts with gpt-4o-mini (as in the MaAS[1] setting). Multiple issues on the official AFLOW GitHub repository report unstable workflow evolution caused by incorrect Python code, and a recent Google paper [2] also notes that AFLOW only works reliably with certain LLMs.
>
> We report the token utilization for the workflow generation phase below.
>
> | Method | Workflow Generation (average by 5 iteration) | Mermaid → Python Translation (average by 5 iteration) |
> | --- | --- | --- |
> | AFlow | ~8.6k tokens | - |
> | **MermaidFlow** | **~8.4k tokens = ~7k for generating workflow candidates + ~1.4k for the LLM to choose which workflow to execute** | **~2k** |
> | **MermaidFlow (with rule-based Mermaid→Python transformer)** | **~8.4k tokens** | - |
>
> We don’t report the evaluation cost because it varies significantly across workflows and is therefore not directly comparable, whereas the workflow generation cost is much more informative. At the per-iteration level, our method does not have an advantage, even with a more lightweight representation, because we still require a Mermaid→Python transformation; **in this paper, that transformation is implemented with an LLM, which incurs additional cost. After submitting the paper, we are more aware of this issue and therefore** **implemented a rule-based Mermaid→Python converter.** Thanks to Mermaid’s graph-like representation, we can easily parse Mermaid into NetworkX (a Python graph library) and apply a set of rules to transform Mermaid workflows into LangGraph programs. **We have included this tool in the anonymous link below and prepared several demos for you to try.** We also plan to publicly release this tool to further support Mermaid as a workflow representation.
>
> Here is the tool repo: https://anonymous.4open.science/r/mermaid2python-E42F/

---

> ### Author Response · Authors · 2025-11-21
> **Response to Reviewer 3qM1 (3/4)**
>
> > **Weaknesses 3.1: Unclear role of the Checker:** If Mermaid provides strong type constraints, the checker should rarely trigger, as LLMs have very low probability of generating syntactically invalid Python code. However, the checker is actually a critical component in MermaidFlow (Section A.2), suggesting that the constraints are not strong enough. A >90% success rate still means 10% of generations require retry, which remains quite high.
> >
>
>
>
> The checker is a crucial component in our design. **We cannot rely solely on LLMs to consistently produce correct workflows; explicit mechanisms are needed to ensure that generated workflows satisfy predefined structural rules. Mermaid not only allows us to verify that a script adheres to Mermaid’s grammar, but its graph-based representation also enables structural checks to ensure that the workflow topology follows our design principles.** On the Mermaid side, we can therefore guarantee that all scripts are valid. However, in this paper we still rely on an LLM to translate Mermaid into executable Python, and the remaining ~10% failure cases arise entirely from this translation step. We fully agree that, from a safety perspective, a 10% failure rate is not acceptable. To address this, we have developed a new version that uses rule-based transformation from Mermaid to Python (implemented on top of LangGraph). As part of our effort to make solid, long-term contributions in this direction, we have already shared the corresponding code in our response to Weakness 2.2 and will publish it in the future.
>
> > **Weaknesses 3.2: Incomplete operator definitions:** The paper claims "safety-constrained", but specific safety properties (such as deadlock-freedom, termination) are not defined. There is also a lack of case study analysis on how directly generating Python code workflows violates these safety properties.
> >
>
> **The term “safety-constrained” in our work primarily applies at the Mermaid script level.** Mermaid is a well-studied language, and LLMs already possess substantial prior knowledge about how to use it. However, its original purpose is general graph representation rather than workflow specification. Therefore, to obtain executable workflows, **we must impose additional constraints, for example, enforcing a single entry node and a single exit node, and requiring that certain node types can only be activated under specific rules.** In this paper, we define five rules for constructing workflows, as listed in Appendix A.2. As shown in Question 1.2, our checker can detect violations of these rules and repair the workflows into valid ones.
>
> > **Weaknesses 4: Conceptually flawed taxonomy**
> >
>
> Thank you for pointing out this definition. We revise this section and change the workflow type to non-agentic reasoning workflow.
>
> > **Questions 1.1: Mermaid vs. Python comparison**: Show side-by-side examples where (a) Python-based workflow generation fails but Mermaid succeeds, and (b) demonstrate what specific safety properties are violated in the Python case (e.g., deadlock, non-termination, type mismatch).
> >
>
> We present several concrete failure cases of Python-based workflows in our paper; more detailed examples can be found in **Appendix C**. These failures can be grouped into the following main categories:
>
> 1. **Python’s flexibility often misleads the LLM into producing superficially reasonable but unreliable control logic**, such as:
>     1. Unreliable if conditions, where the LLM’s decision on whether a condition should trigger is incorrect, since we cannot guarantee the behavior of the LLM’s output.
>     2. Using a single prompt to drive a for loop over many iterations, which offers essentially no benefit when the temperature is set to 0.
> 2. **Incorrect instance initialization**, a common issue when an LLM encounters a new repository:
>     1. Parameters may be incorrect or missing.
> 3. **Importing or referencing nonexistent modules or repositories**.
>
> Most failures stem from LLMs struggling to reliably generate correct Python scripts, so Mermaid’s advantage is partly inherent. Using Mermaid as an intermediate representation mitigates these failure modes in Python-based workflows: its graphical, lightweight syntax and widespread use as pseudocode on platforms like GitHub make it easier for LLMs to produce reliable implementations conditioned on a Mermaid diagram (likely because such patterns are well represented in their training data). Nevertheless, our original LLM-based Mermaid→Python translation still has roughly a 10% chance of producing incorrect code. With our new rule-based mermaid2python tool, which deterministically maps Mermaid scripts to Python, we can guarantee that the generated Python code is correct whenever the underlying Mermaid workflow is valid.

---

> ### Author Response · Authors · 2025-11-21
> **Response to Reviewer 3qM1 (4/4)**
>
> > **Questions 1.2:** **Safety constraint effectiveness**: Provide concrete examples showing how safety constraints improve efficiency or effectiveness during evolution. What specific invalid mutations are prevented? How much retry overhead is avoided?
> >
>
> **In the submitted version of the paper, we enforce only five constraints. In practice, LLMs follow Mermaid quite well: among these constraints, only W2, W4, and W5 are occasionally violated, and all such violations are detected by the checker, thanks to Mermaid’s structured representation.**
>
> The table below summarizes the violation counts across four benchmarks, each run for 20 iterations. For more detailed examples, please refer to **Appendix D** of our paper.
>
> |  | W1: Ensure PROBLEM and RETURN nodes exist. | W2: No isolated nodes are allowed. | W3: Inputs and outputs must be of Interface type. | W4: Do not create new types. | W5: Ensemble nodes must have at least two inputs |
> | --- | --- | --- | --- | --- | --- |
> | **Violation counts over 20×4 rounds** | 0 | 4 | 0 | 2 | 3 |
>
> > **Questions 1.3: Checker analysis**: Given that the checker has >90% success rate (implying ~10% retry), provide examples of the 10% failed cases. What constraint violations occur? Why doesn't Mermaid's type system prevent these?
> >
>
> **Actually, all of the observed failure cases are caused by the `Mermaid→Python` transformation process.** On the Mermaid side, thanks to the checker, the system rarely produces invalid workflows. **However, when using an LLM to translate Mermaid into Python, failures can still occur, including several of the issues described in Question 1.1.** To address this limitation, we have implemented a rule-based tool to convert Mermaid scripts into Python. This not only reduces token consumption but also lowers the error rate. We have included this transformer in our response in Weaknesses 2.2.
>
> This part should be combined with Question 1.1, using Mermaid as an intermediate representation helps mitigating these problems. **When the LLM generates Python code conditioned on an existing Mermaid workflow, it receives a much clearer structural blueprint, so the resulting code is more disciplined and conservative in its behavior, leading to a higher success rate.** Although this mechanism is difficult to formalize, empirically we observe that providing the corresponding Mermaid script before asking the LLM to produce Python tends to yield more reliable code. A reasonable explanation is that Mermaid is often used as a form of pseudocode, and platforms like GitHub natively support Mermaid rendering (e.g., in issues or README files), so LLMs have learned to map Mermaid diagrams into stable Python implementations.
>
> > **Questions 1.4: Component reuse**: Demonstrate how MermaidFlow enables reusing good components (e.g., effective prompts) from suboptimal workflows, addressing the AFlow limitation you mentioned. Show concrete examples from your evolution process.
> >
>
> In the evolution process, MermaidFlow can stably increase or delete node from the father workflow. Which means it will reuse the most part of father workflow. In this paper, we follow the same setting as AFLOW by using one workflow to solve all the questions. In such setting the module reuse is very hard but use the most promising workflow part would be a good choose. We have explore some new method by query-base, which could retrieve related mermaid script by graph similarity. This is kinda of future work by exploiting mermaid representation.
>
> [1] Zhang, G., Niu, L., Fang, J., Wang, K., Bai, L. and Wang, X., 2025. Multi-agent architecture search via agentic supernet. arXiv preprint arXiv:2502.04180.
>
> [2] Zhou, H., Wan, X., Sun, R., Palangi, H., Iqbal, S., Vulić, I., Korhonen, A. and Arık, S.Ö., 2025. Multi-agent design: Optimizing agents with better prompts and topologies. *arXiv preprint arXiv:2502.02533*.
>
> ---
>
> **We sincerely appreciate Reviewer 3qM1’s careful review and thoughtful feedback. Reviewer 3qM1 not only raised highly valuable questions, but these questions also strike at the key components of our method. This made us truly feel the depth of your careful reading and thoughtful engagement with our paper, for which we are very grateful. Your comments also demonstrated that you are an expert in this field, and we feel honored to have the opportunity to engage in deeper discussion with you. More importantly, many of your insights align closely with our own, for example, your suggestions on improving the reliability of Python script generation (90% success rate is not enough), which has been our main focus of improvement after the submission. We hope that the additional materials we provide can offer you more useful and meaningful information. If you think the response adequately addressed your concerns, please adjust the score accordingly.**

---

> ### Comment · Reviewer_3qM1 · 2025-11-24
>
> First, I would like to commend the pursuit of generation process stability demonstrated in your work, and the specially developed rule-based stable and efficient toolkit for this, which is indeed praiseworthy. **Following this, I wish to elaborate on the roles that structural stability and instability play in workflow evolution, how these properties are manifested in your current method and results, and where genuine improvements might lie or to what extent they can be achieved.**
>
> Regarding the significance of the stability achieved in your work, if the primary contribution is merely avoiding the types of errors listed in Appendix C, I must gently note that, **based on my personal observation**, modern efficient models like **gpt-4o-mini** have largely improved to the point where such rudimentary errors occur infrequently. Moreover, even when such errors do occur, they would typically be filtered out by the search algorithm during the evolutionary process.
>
> To illustrate what would constitute a **truly advanced structural optimization**, consider the concept of **Strategy Branching**. For example, in the HotpotQA benchmark, although all questions are packaged together, solving them effectively requires "overfitting" to specific scenarios with empirical strategies. These range from basic official classifications like 'comparison' and 'bridge', to more granular strategies such as union vs. intersection in comparison, or 2-hop vs. 3-hop reasoning in bridge questions. These strategies can be deceptive. While it is possible to let an LLM handle both strategy selection and execution in a single call, the theoretical upper bound for performance is lower than a structured "classify-then-select" approach. However, such sophisticated structures are extremely difficult to emerge in current workflow generation methods. This is because learning these structures is a long-term process requiring accumulated experience, whereas current methods rely on feedback from single simulations which offer only limited, shallow case reflection.
>
> this creates a contradiction: workflows with incomplete strategy branches cannot yield good results. this limitation is evident in the optimal workflows provided in your appendix b, where the ensemble branches consist of vague, high-level prompts rather than strategies relevant to specific problem subdivisions. **although your method claims to bring 'safety-constrained evolution', the lack of such strong structural analysis (as illustrated by the strategy branching example) makes this claim unconvincing.** the current approach seems unable to bridge the gap between simple workflows and complex, experience-based structures.
>
> Furthermore, a fundamental distinction between workflow generation and traditional model training lies in **interpretability**. Unlike model weights—opaque matrices where the significance of specific values is indecipherable—a high-quality workflow should be transparent and understandable. This interpretability presents both a challenge and an opportunity. However, upon examining the 'optimal workflows' provided in Appendix B, I find them lacking in structural novelty. I see no evidence of advanced structures such as **test case augmentation** in code generation, or **strategy branching and selection gates** (along with potential reflection/re-selection loops) in reasoning tasks. Similarly, the prompt improvements appear minimal, lacking the strong, rule-based instructions often necessary for multi-hop datasets (I apologize that I cannot provide specific examples as this comes from my personal accumulated experience). Consequently, the generated workflows seem to offer little advantage over direct prompt optimization, failing to leverage the unique opportunity for interpretable structural innovation that this field offers.
>
> In conclusion, I cannot provide a higher rating.

---

> ### Author Response · Authors · 2025-11-27
> **Official Comment by Authors (1/2)**
>
> **We appreciate your comments and the insights you have shared with us.** To ensure we address your concerns accurately, we understand them as twofold: (1) why it is necessary to emphasize update stability and the use of a checker, and (2) why our paper does not demonstrate more structurally novel workflows. More generally, you are asking whether the Mermaid representation is truly necessary. We would be glad to discuss these aspects further.
>
> There has been substantial discussion on how to reuse modules or fine-tune models to generate workflows that achieve higher accuracy on specific benchmarks. Python is indeed a reasonable choice for representing workflows, as it leverages the full capability of LLMs. **However, recent systems often prioritize accuracy performance while relaxing structural correctness, but this leads to significant validity issues.** Many existing methods, even AFLOW, attempt to introduce such constraints through meta-prompts (as noted in our response to Weakness 1.1). **Yet it remains unclear how these constraints can actually be enforced.** In our view, if we require that the generated workflow **MUST** satisfy these constraints strictly, then **more than half** of the workflows produced by AFLOW become **INVALID!!** and **AFLOW itself is unable to detect these violations even.** This demonstrates that workflow systems require an explicit mechanism for verifying their own structural correctness.
>
> From this perspective, our solution is to adopt a new workflow representation, **one that LLMs can easily understand and generate, and, most importantly, which can be reliably verified.** A workflow is **mathematically a graph**, which implies that its constraints and search space should naturally be defined **within the domain of graph structures**. Therefore, it is reasonable to use a language explicitly designed for describing graphs. In our case, **we choose Mermaid because LLMs handle it well, it offers clear visualization, and, critically, it is easy to verify.**
>
> To further illustrate the importance of enforceable constraints, we conducted additional experiments using a topological constraint that may not directly affect final accuracy but is highly relevant in real-world workflow design, such as controlling inference cost. Specifically, we constrain the number of nodes and edges to remain within predefined limits. We inject this constraint into the prompt and evaluate the validity rate of generated workflows under this topological constraint, as shown below:
>
> |  | AFLOW | MermaidFlow w/o checker | MermaidFlow w/ checker |
> | --- | --- | --- | --- |
> | topological constraint | 25.4% | **80.4%** | **100%** |
>
> As the table indicates, using an appropriate representation significantly improves the success rate of generating valid workflows. Moreover, incorporating **a verification mechanism ensures that every generated workflow strictly satisfies the required constraints**. **Mermaid naturally supports such verification,** whereas Python-based workflow scripts offer no comparable guarantees.

---

> ### Author Response · Authors · 2025-11-27
> **Official Comment by Authors (2/2)**
>
> As you noted, the final workflow in the paper appears general and relies on high-level prompts. This is largely a consequence of our experimental setup and the predefined node types. However, similar to AFLOW, our setting requires producing a single workflow that must handle all questions in the dataset. **During evolution, the system does generate workflows with deeper structures and greater complexity, but these are eventually compressed when the workflow is forced to generalize across the entire dataset. We could consider about using strategy branching,** If we were to introduce additional node types, such as a router node, the framework could readily support combining multiple workflows via node–edge composition (this action is natural in Mermaid) and routing-based selection.
>
> When we switch to a query-based workflow setting, **where each query is allowed its own workflow, the evolution becomes much more expressive.** In this setting, we can apply reflection-based refinement across several rounds, enabling each workflow to iteratively improve until it can reliably solve its target query. We are currently applying this prototype to ARC, and we are happy to share several examples demonstrating the potential of stability-based updates in producing deeper and more problem-specific workflows.
>
> Here is several examples in ARC benchmark, listed in this anonymous link: https://anonymous.4open.science/r/query-base-C320/. In the workflow evolution shown in this link, the workflows evolve in a stable manner: each new workflow preserves the core structure and prompt of its predecessor while growing deeper and more interconnected. This ensures that the final-answer LLM receives sufficiently rich information accumulated from previous workflow steps.
>
> ---
>
> We would like to emphasize that **future workflow representations must incorporate explicit, machine-checkable verification mechanisms to guarantee validity, rather than relying solely on LLM behavior or informal prompt constraints. Designing workflows that rigorously satisfy given rules is a key problem in the direction of automatic workflow generation.**
>
> We thank the reviewer again for the time devoted to evaluating our rebuttal responses and for enriching the insightful dialogue. We genuinely appreciate the opportunity to learn from your perspective, and your comments have helped us sharpen both the framing and future direction of this line of research. With best wishes, we look forward to continued constructive engagement.

---

### Meta-Review · Area_Chair_gKxa · 2026-01-06

**Summary:**

This paper proposes a framework that utilizes the Mermaid graph language and evolutionary programming to generate verifiable agentic workflows. The decision to reject is driven by the marginal performance gains which appear to be partially exaggerated by unfair experimental comparisons (unequal ensemble voting sizes), and the omission of critical related work (e.g., EvoFlow, EvoAgentX, EVOAGENT, EvoPlan). Without benchmarking against these highly relevant evolutionary baselines (some of which seemingly outperform the proposed method), the specific contribution and necessity of the Mermaid-based approach remain unproven.

Therefore, for the benefit of this paper, we regretfully recommend rejection. Note that this is not a discouragement. The authors are encouraged to address these concerns, and we believe the paper has the potential to become a strong future submission.

**Reviewer Concerns:**

While the authors addressed concerns about the curriculum mechanism, assumptions on information gain, and instruction following, some major concerns still remain. Specifically:

- Missing Related Work & Performance Baselines (Reviewer cd1T): A significant number of highly relevant prior and concurrent works utilizing evolutionary algorithms for agentic workflows were neither cited nor compared. Specifically, EvoFlow, EvoAgentX, EVOAGENT, and EvoPlan. The paper's performance appears lower than some off these established methods, and without a detailed comparison, it is difficult to assess the actual contribution of using Mermaid-based evolution over these existing state-of-the-art evolutionary frameworks.

- Experimental Fairness (Reviewer 3qM1): A critical discrepancy in the evaluation setup remains a major issue. The authors admitted that MermaidFlow used a 5-sample voting ensemble while the baseline AFlow used only 3-sample voting. Although the authors provided a new 3-sample result in the rebuttal (55.14% accuracy), this performance is lower than the reported main result and much closer to the baseline (52.81%), confirming that the original performance gap was inflated by the ensemble size difference.

- Marginal Gains and Novelty (Reviewer 3qM1): The performance improvement over AFlow is marginal (~2%), and the method is viewed as an incremental variant of existing evolutionary paradigms rather than a fundamental breakthrough. The necessity of the Mermaid representation is not fully justified.

**Reviewer Scores:**

The reviewers are likely to maintain the current score.

---

### Decision · Program_Chairs · 2026-01-26

Reject